# PATCHCAT: RETHINKING TEMPORAL TOKENIZATION IN TIME SERIES FORECASTING

## ABSTRACT

Temporal tokenization serves as a fundamental component in time series forecasting, transforming raw signals into token representations. Existing temporal tokenizers fall into three typical categories, mapping time series into tokens at the point-wise, patch-wise, or variable-wise level. Through a fair comparison, we observe that none of these paradigms simultaneously balance forecasting accuracy with computational efficiency. We point out the key to achieving both accuracy and efficiency lies in using a single token to preserve global sequence information and local semantics of temporal mutations. Therefore, we propose PatchCat, which segments the input time series into consecutive patches and concatenates these embeddings in chronological order. This workflow not only preserves local semantics and sequential information, but also enables univariate series to be compressed into a single token, achieving efficiency comparable to variable-wise methods. To further enhance representational capacity, we adopt a linearly increasing dimension allocation strategy and the variable-wise affine transformations. Experiments show that replacing the tokenizer in many existing methods with PatchCat can effectively improve prediction performance. To further leverage PatchCat's strengths, we develop PCMLP, a simple yet powerful model based on a multilayer perceptron. Extensive experiments across 13 challenging real-world datasets demonstrate that our approach achieves competitive performance compared to state-of-the-art methods.

## 1 INTRODUCTION

Time series forecasting (TSF) is a fundamental problem in machine learning, with wide-ranging applications in weather prediction, traffic planning, financial risk management, and energy systems (Gao et al., 2023; Han et al., 2024b; Bi et al., 2023). Despite recent advancements enabled by deep learning techniques (Jia et al., 2024; Chen et al., 2024), forecasting long and complex time series remains challenging. A common difficulty lies in how to transform raw temporal signals into effective representations that capture both fine-grained local patterns and global dependencies while remaining computationally efficient. This transformation is typically handled by the *temporal tokenizer*, which converts raw time series into tokens for downstream backbones. Therefore, the design of a tokenizer is more than just a preprocessing step; the quality and quantity of its output tokens impact the overall model's forecast accuracy and computational efficiency, making it a critical component in the modern time series forecasting methods.

Thus, the design of temporal tokenizers has become a rising topic in the TSF community. Most deep learning-based models, such as Informer (Zhou et al., 2021) and TimeMoE (Shi et al., 2024), use a *point-wise* tokenization strategy, treating each time point as an individual token. While this strategy ensures the completeness of temporal information, it lacks local semantic information in time series and incurs high computational cost in long-term forecasting tasks. To alleviate these limitations, later works (Zhang & Yan, 2023; Nie et al., 2023; Tang & Zhang, 2025) adopt a *patch-wise* strategy, dividing each univariate sequence into multiple (possibly overlapping) patches. This approach enriches local contextual understanding while reducing the overall number of tokens. Recently, a *variable-wise* strategy (Liu et al., 2024a) has been proposed, transforming each univariate time series into a single token. This extreme reduction of tokens offers substantial efficiency of the overall forecasting process, albeit at the expense of fine-grained temporal information. An open question thus

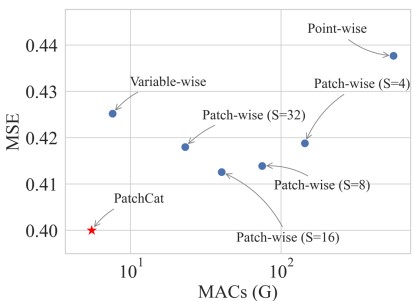 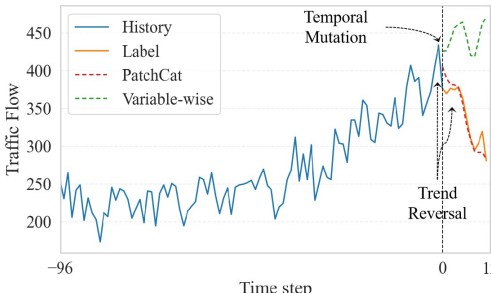

Figure 1: Compare performance and efficiency of tokenizers on the same backbone.

Figure 2: An example on the PEMS08 dataset: temporal mutation causes a trend reversal in traffic flow.

arises: *Is there a temporal tokenizer that can simultaneously achieve both high predictive accuracy and efficiency, serving as a silver bullet for time series forecasting?*

Direct comparison of the above three kinds of temporal tokenizers across existing works is confounded by the usage of heterogeneous backbones. To enable a fair evaluation, we first summarize these tokenization strategies from the perspective of patch size: point-wise corresponds to a patch size of one, and variable-wise corresponds to a patch size equal to the input length. Therefore, we can use the patch-wise tokenizer as a unified version of all three tokenizers, reducing the unfairness caused by implementation. Then, we use multi-layer Transformers (Vaswani et al., 2017) as an encoder to mine temporal correlation from tokens. Thereafter, a linear projection is used to transform the latent representation into the predictions. More details can be viewed in Appendix A. Under this unified *tokenizer-encoder-projection* pipeline, we fairly obtain the prediction accuracy and computational cost of these three types of tokenizers. As shown in Figure 1, the patch-wise tokenizer achieves the lowest forecasting error by aggregating local time steps into semantically meaningful patches, while the variable-wise tokenizer offers the highest efficiency by compressing each univariate into a single token.

How can variable-wise tokenizers achieve performance comparable to patch-wise tokenizers while maintaining their superior speed? We argue that the main challenge lies in addressing two inherent weaknesses of compressing an entire time series into a single token: such approach struggles to preserve **temporal order information** and inevitably over-smooths **temporal mutations**, leaving the tokenizer insensitive to recent temporal mutations as shown in Figure 2. To overcome these limitations, we propose PatchCat, a novel temporal tokenizer that assists forecasting models in achieving advanced prediction accuracy with high efficiency. PatchCat transforms consecutive time patches into embeddings and then concatenates these embeddings in chronological order to form a single token. It centers on two core operations: (i) the **patch-then-concat** strategy, which effectively ensures both local semantics retention and the maintenance of sequential information within a compact token. (ii) the **linearly increasing dimension allocation** that assigns more embedding space to recent patches, reflecting the *recent matters most* principle and capturing temporal mutations. Moreover, PatchCat can be used to improve the forecasting performance of many existing methods by replacing the original tokenizer. To better leverage PatchCat's strengths, we combine it with multi-layer perceptron (MLP) to form a simple and efficient model, PCMLP. Experiments show that our method achieves the best performance on 13 real-world datasets over eight advanced methods.

In summary, this paper contributes:

- We summarize existing temporal tokenizers into point-wise, patch-wise, and variable-wise strategies, and provide a fair empirical comparison. Our analysis identifies the key challenge of efficient tokenization as encoding local semantics within the single token.

- We propose PatchCat, an efficient temporal tokenizer based on *patch-then-concat*, which embeds local semantics and sequential structures into one token. PatchCat enhances the performance of multiple existing time series forecasting models.

- We combine PatchCat with MLP as PCMLP, which is a simple yet efficient method. PCMLP achieves consistent state-of-the-art performance across 11 real-world datasets and offers significant efficiency advantages.

## 2 RELATED WORK

Recently, TSF research has flourished, with a variety of methods with different architectures emerging (Lin et al., 2023; Liu et al., 2022; Kong et al., 2025). Among these methods, the temporal tokenizer is a key common component, which can be categorized into three types. First is the point-wise tokenization, which has been used in early time series forecasting methods (Box et al., 2015) and is also widely used in the latest deep-learning-based models (Shi et al., 2024). It treats each time point as an individual token to ensure the completeness of temporal information. Recently, some studies (Nie et al., 2023; Cheng, 2024) suggest that time series have patch-wise local semantics, aggregating information from multiple time points can better reflect temporal changes and is less susceptible to the influence of outliers. Therefore, they propose a patch-wise tokenizer that divides the time series into consecutive, potentially overlapping, patches and treats each patch as a token. However, the number of tokens used by these two methods increases linearly with the length of the input time series, which imposes significant computational overhead in long-term TSF tasks on sophisticated forecasting backbones. iTransformer (Liu et al., 2024a) takes an inverted perspective, transforming each variable into a token. We refer to this as the variable-wise tokenizer, which has the highest efficiency but lacks the fine-grained information about the time series.

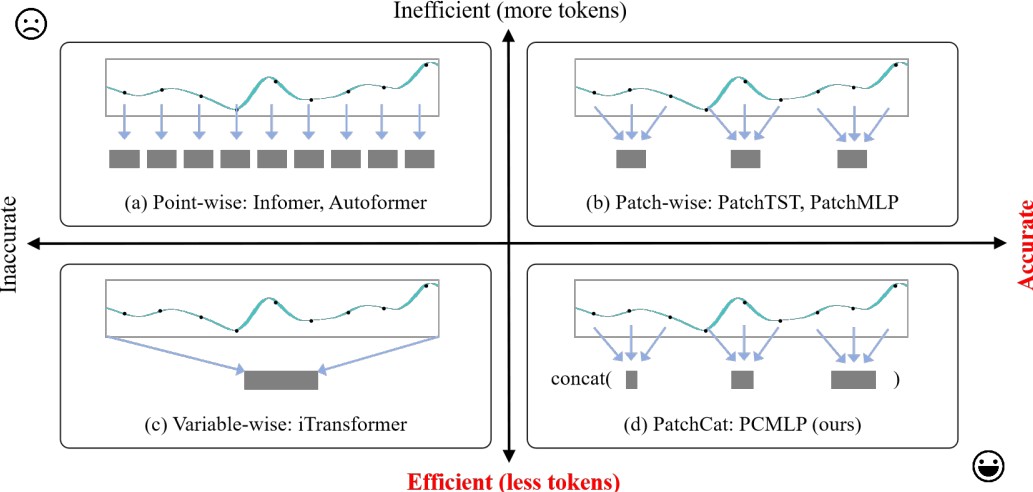

Figure 3: Summary of four kinds of temporal tokenizers.

As shown in Figure 3, we establish a connection between the above three types of temporal tokenizers from the perspective of patch size: the point-wise (variable-wise) tokenizer can be viewed as a special case of patch-wise tokenizers with patch size equal to 1 (patch size equal to the input length). With the patch size changes, the semantics of tokens shift between local and global, and the number of tokens also changes. Unlike previous works, our proposed PatchCat uses a **patch-then-concat** solution that combines high performance and high efficiency.

## 3 METHOD

In time series forecasting, given the historical observations with $C$ variables during $L$ time steps $\mathcal{X} = \{X_1, X_2, \cdots, X_C\} \in \mathbb{R}^{L \times C}$, we predict the future $H$ time steps $\mathcal{Y} = \{Y_1, Y_2, \cdots, Y_C\} \in \mathbb{R}^{H \times C}$. Mathematically, we represent this process as:

$$\psi\left(\phi\left(\mathcal{X}_{t-L+1:t} \in \mathbb{R}^{L \times C}\right)\right) \longrightarrow \hat{\mathcal{Y}}_{t+1:t+H} \in \mathbb{R}^{H \times C}, \tag{1}$$

where $\phi(\cdot)$ is the temporal tokenizer and $\psi(\cdot)$ is the backbone network. To efficiently convert raw time series into deep embeddings, we propose a novel temporal tagger, PatchCat. For $X_i$ where $i \in \{1, 2, \cdots, C\}$, PatchCat converts patches into embeddings using linear layers with varying output dimensions and concatenates these embeddings in time order to obtain a single token. Leveraging PatchCat's powerful representational capabilities, we achieve leading prediction accuracy and speed using a simple multilayer perceptron (MLP) as the backbone model.

## 3.1 PATCHCAT: AN EFFICIENT TEMPORAL TOKENIZER

The quality and quantity of tokens determine the prediction accuracy and computational overhead. Given the excellent performance of patch-wise tokenization and variable-wise tokenization on these two metrics, we point out that the key challenge of efficient temporal tokenization lies in how to capture local semantics in time series under the constraint of a very small number of tokens. Therefore, we propose a **patch-then-concat** tokenization strategy, PatchCat, which projects patchified time series into embeddings and then concatenates them into a single token in chronological order. This approach effectively preserves the local semantics of time series and the sequential relationships between the local semantics using only one token.

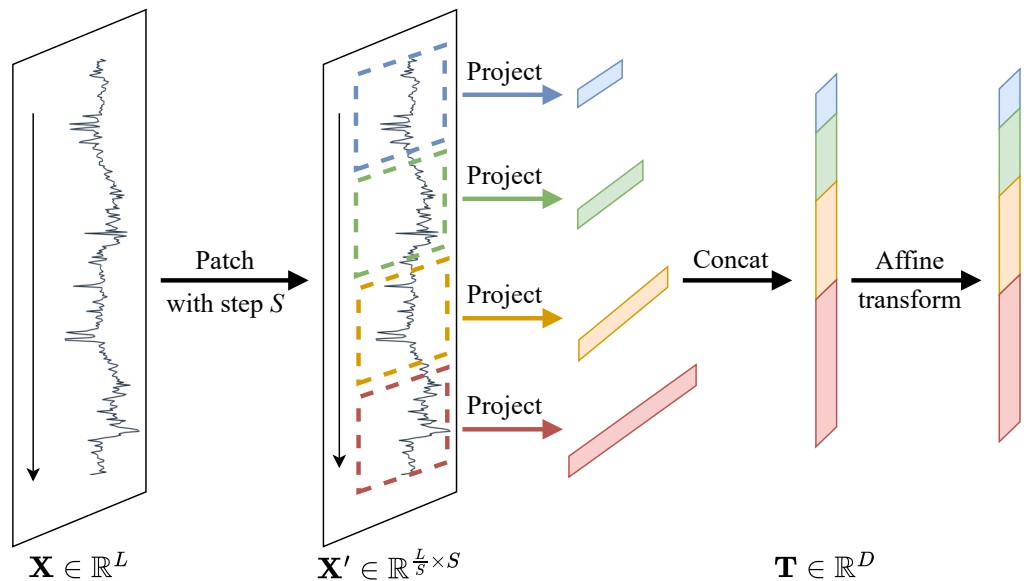

Figure 4: PatchCat architecture. The time series is segmented into patches; then transformed into embeddings, with a linearly increasing dimension allocation strategy; all embeddings are concatenated in chronological order; and an affine transformation is performed on the variable dimensions.

As shown in Figure 4, PatchCat consists of four sequential steps: patching, projection, concatenation, and transformation. Take the tokenization process of a univariate time series $X$ as an example. First, we partition raw time series $X \in \mathbb{R}^L$ into contiguous patches $X' \in \mathbb{R}^{\frac{L}{S} \times S}$ with patch size $S$. Each patch contains multiple consecutive time points, reflecting local temporal patterns such as rise, fall, and oscillation. Then, we apply the learnable linear layers to project $X'$ into embeddings. Subsequently, we concatenate the embeddings in chronological order to obtain one token $\mathbf{t} \in \mathbb{R}^D$ with dimension $D$ for the entire time series as:

$$\mathbf{t} = \text{Cat}\left(\theta_1(X'_1), \theta_2(X'_2), \cdots, \theta_{\frac{L}{S}}(X'_{\frac{L}{S}})\right), \tag{2}$$

where $\theta$ is the linear layer and $\text{Cat}(\cdot)$ is the concatenation operation. After that, PatchCat yields one token per variable, encapsulating the ordered local semantics into the single token. For multivariate time series, learnable parameters of projection are shared across variables. Because similar time series data may have different semantics across variables, shared parameters may lead to polysemy in tokens, making prediction more difficult. Therefore, we apply an affine transformation on multivariate tokens $\mathbf{T} \in \mathbb{R}^{C \times D}$ along the variable dimension to enhance inter-variable token distinguishability:

$$\mathbf{T} = [\mathbf{t}_1, \mathbf{t}_2, \cdots, \mathbf{t}_c] \times \alpha + \beta, \tag{3}$$

where $\alpha$ is initialized to an all-ones vector, and $\beta$ is initialized to an all-zeros vector.

There are two noteworthy details of PatchCat: Firstly, **no overlap during patching**. Some patchwise tokenizers (Nie et al., 2023) required overlap between neighbor patches. However, PatchCat concatenates the patch embeddings to alleviate this semantic fragmentation, eliminating the need for

overlap. No-overlap operation slightly reduces computational overhead. Secondly, **the dimension allocation strategy** in projection. Based on a simple prior: recent patches have a greater impact on future predictions (recent matters most), we use a linearly increasing dimension allocation strategy, which means that we assign larger dimensions to the nearer patch embedding. Assuming the dimension of the first patch embedding is $d$, the dimension of the $k$-th patch embedding is $k \times d$, and the token dimension $D = \frac{1 + \frac{L}{S}}{2} \times d$. We provide a theoretical analysis of the computational complexity in the Appendix B.

## 3.2 PCMLP: A SIMPLE TSF MODEL BASED ON PATCHCAT

Considering that PatchCat converts time series into high-quality tokens, we propose a simple MLP-based time series forecasting model, PCMLP. As shown in Figure 5, PCMLP includes time encoding, PatchCat, multi-layer perceptron, output projection, and instance normalization (and de-normalization).

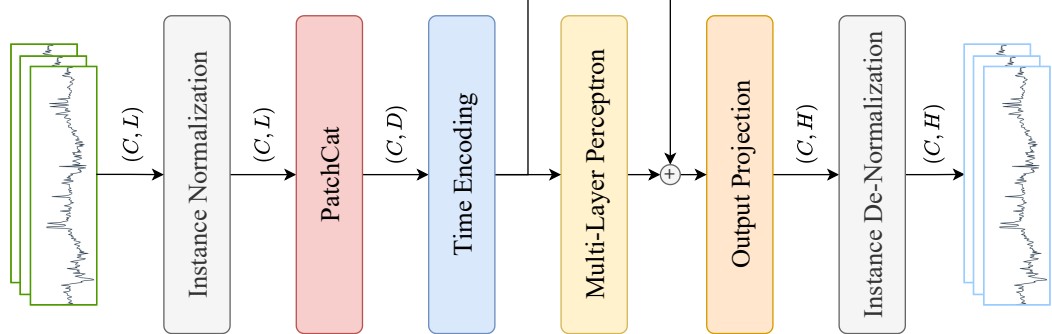

Figure 5: The pipeline of PCMLP. The raw data $\mathcal{X} \in \mathbb{R}^{C \times L}$ passes through six lightweight modules in sequence to obtain the prediction results of $\hat{\mathcal{Y}} \in \mathbb{R}^{C \times H}$.

**Time encoding.** In time series, each time point corresponds to a unique timestamp, which contains rich temporal information, such as year, month, day, hour, and minute. This temporal feature reflects temporal patterns, especially those related to periodicity. For example, if a time series has a daily cycle, the patterns of temporal change at the same moment are roughly similar. To incorporate time features into the model, we draw on a similar approach to STID (Shao et al., 2022), which maps timestamps into a learnable time encoding shared across variables. Since most data has a daily or weekly cycle, we select two time features: the time point of day (tod) and the day of week (dow). We add tokens to the time encodings $\mathbf{E}_{\text{tod}}$ and $\mathbf{E}_{\text{dow}}$ to obtain more representational tokens $\mathbf{E} \in \mathbb{R}^{C \times D}$ with absolute time information:

$$\mathbf{E} = \mathbf{T} + \mathbf{E}_{\text{tod}} + \mathbf{E}_{\text{dow}}. \tag{4}$$

**Multi-layer perceptron and output projection.** We use a multi-layer perceptron (MLP) network as an encoder, which consists of two fully connected layers (Linear) interspersed with GeLU activations. And we add a residual connection and layer normalization between the token $\mathbf{E}$ and the MLP's output. The whole MLP encoder is defined as:

$$\mathbf{H} = \mathbf{E} + \text{LayerNorm}\left(\text{MLP}\left(\mathbf{E}\right)\right), \quad \text{MLP}\left(\mathbf{E}\right) = \text{Linear}\left(\text{GeLU}\left(\text{Linear}\left(\mathbf{E}\right)\right)\right). \tag{5}$$

The output $\mathbf{H}$ is fed into a linear layer with dropout operation, projecting the learned hidden representations onto the target forecasting horizon:

$$\hat{\mathcal{Y}} = \text{Linear}\left(\text{Dropout}\left(\mathbf{H}\right)\right). \tag{6}$$

**Instance normalization and de-normalization.** Time series data often exhibit distribution shifts between training and testing sets. Recent works have pointed out that using normalization strategies between the input and output of models can alleviate this problem. We adopt a simple yet effective instance normalization and de-normalization method used in previous works (Liu et al., 2024a; Lin et al., 2024), which removes the mean and variance of the input data before and after the model.

Table 1: Long-term multivariate time series forecasting results with prediction lengths $H \in \{12, 24, 48, 96\}$ for PEMS and $H \in \{96, 192, 336, 720\}$ for others and lookback length $L = 96$. Results are averaged from all prediction lengths. Avg means further averaged by subsets. The best is in **red** and the second is underlined. Full results are listed in Appendix D.1.

| Models | PCMLP (Ours) | | TQNet (2025) | | TimeXer (2024) | | CycleNet (2024) | | iTransformer (2024a) | | MSGNet (2024) | | TimesNet (2023) | | PatchTST (2023) | | DLinear (2023) | |
|---|---|---|---|---|---|---|---|---|---|---|---|---|---|---|---|---|---|---|
| Metric | MSE | MAE | MSE | MAE | MSE | MAE | MSE | MAE | MSE | MAE | MSE | MAE | MSE | MAE | MSE | MAE | MSE | MAE |
| ETT(Avg) | **0.365** | **0.384** | 0.368 | 0.388 | **0.365** | 0.388 | 0.373 | 0.390 | 0.383 | 0.399 | 0.389 | 0.406 | 0.391 | 0.404 | 0.381 | 0.397 | 0.442 | 0.444 |
| ECL | **0.163** | **0.252** | 0.164 | 0.259 | 0.171 | 0.270 | 0.168 | 0.259 | 0.178 | 0.270 | 0.194 | 0.301 | 0.193 | 0.295 | 0.205 | 0.290 | 0.212 | 0.300 |
| Traffic | **0.425** | **0.259** | 0.445 | 0.276 | 0.466 | 0.287 | 0.472 | 0.301 | 0.428 | 0.282 | 0.660 | 0.382 | 0.620 | 0.336 | 0.481 | 0.300 | 0.625 | 0.383 |
| Weather | **0.238** | **0.263** | 0.242 | 0.269 | 0.241 | 0.271 | 0.243 | 0.271 | 0.258 | 0.278 | 0.249 | 0.278 | 0.259 | 0.287 | 0.259 | 0.273 | 0.265 | 0.317 |
| PEMS03 | **0.097** | **0.193** | 0.097 | 0.203 | 0.112 | 0.214 | 0.118 | 0.226 | 0.113 | 0.221 | 0.150 | 0.251 | 0.147 | 0.248 | 0.180 | 0.291 | 0.278 | 0.375 |
| PEMS04 | **0.085** | **0.185** | 0.091 | 0.197 | 0.105 | 0.209 | 0.119 | 0.232 | 0.111 | 0.221 | 0.122 | 0.239 | 0.129 | 0.241 | 0.195 | 0.307 | 0.295 | 0.388 |
| PEMS07 | **0.072** | **0.162** | 0.075 | 0.171 | 0.085 | 0.182 | 0.113 | 0.214 | 0.101 | 0.204 | 0.122 | 0.227 | 0.125 | 0.226 | 0.211 | 0.303 | 0.329 | 0.396 |
| PEMS08 | **0.121** | **0.202** | 0.142 | 0.229 | 0.175 | 0.250 | 0.150 | 0.246 | 0.150 | 0.226 | 0.205 | 0.285 | 0.193 | 0.271 | 0.280 | 0.321 | 0.379 | 0.416 |
| Solar | 0.201 | **0.238** | **0.198** | 0.256 | 0.237 | 0.302 | 0.210 | 0.261 | 0.233 | 0.262 | 0.263 | 0.292 | 0.301 | 0.319 | 0.270 | 0.307 | 0.330 | 0.401 |
| BJAQ | **0.452** | **0.435** | 0.462 | 0.442 | 0.466 | 0.448 | 0.459 | 0.439 | 0.482 | 0.455 | 0.456 | 0.448 | 0.481 | 0.458 | 0.460 | 0.444 | 0.456 | 0.481 |

## 4 EXPERIMENT

### 4.1 SETUP

**Datasets.** We conduct experiments on 13 real-world time series forecasting datasets, including ETT series (Zhou et al., 2021) (including four subsets: ETTh1, ETTh2, ETTm1, and ETTm2), Traffic, Weather (Wu et al., 2021), Electricity (Lai et al., 2018) (abbreviated as ECL), Solar-Energy (Solar) (Lai et al., 2018), Beijing Air Quality (BJAQ), and PEMS series (including four subsets: PEMS03, PEMS04, PEMS07, and PEMS08) (Guo et al., 2021). These datasets vary widely in variable number, ranging from a minimum of 7 (ETT series) to a maximum of 862 variables (Traffic), and cover multiple application scenarios such as electricity, transportation, and weather. Consequently, these datasets are sufficient to verify the effectiveness of our proposed method. More details about the above datasets are provided in Appendix C.

**Baselines and metrics.** We compare PCMLP against eight state-of-the-art models, including TQNet (Lin et al., 2025), TimeXer (Wang et al., 2024), CycleNet (Lin et al., 2024), iTransformer (Liu et al., 2024a), MSGNet (Cai et al., 2024), TimesNet (Wu et al., 2023), PatchTST (Nie et al., 2023), and DLinear (Zeng et al., 2023). Following these works, we use the Mean Absolute Error (MAE), the Mean Squared Error (MSE), and the Multiply–Accumulate Operations (MACs) as metrics for comprehensive evaluation.

**Implementation details.** We implement our method based on the open-source time series forecasting library, EasyTSF (Han et al., 2024a). All experiments are conducted on a single NVIDIA GeForce RTX 4090 with 24G memory. We use MSE as the loss function and Adam (Kingma, 2014) as the optimizer. The hyperparameters are selected as follows: the learning rate is searched from $\{0.01, 0.001, 0.0001\}$, the hidden dim $D$ is searched from $\{528, 1056\}$, the dropout rate is set to 0.2 for the ETT series and 0.1 for others. All hyperparameters were tuned solely on the training–validation splits of each dataset, using the validation set for model selection. Code is uploaded in supplementary materials.

### 4.2 PERFORMANCE STUDY

We compare the forecasting errors of PCMLP with eight baseline models on 11 real-world datasets. Lower MSE and MAE values indicate higher forecasting accuracy. The results of our method are

Table 2: Ablation study about PatcCat and PCNet. Using prediction lengths $H \in \{12, 24, 48, 96\}$ for PEMS and $H \in \{96, 192, 336, 720\}$ for others, lookback length $L = 96$ for all datasets. Results are averaged MSE from all prediction lengths and *Avg* is averaged MSE over 11 datasets. The best/second of seven variants is in **red**/underlined. Full results are listed in Appendix D.2.

| Setup | ETT(avg) | ECL | Traffic | Weather | PEMS03 | PEMS04 | PEMS07 | PEMS08 | *Avg* |
|---|---|---|---|---|---|---|---|---|---|
| ① Group | **0.366** | **0.163** | **0.425** | 0.238 | 0.112 | 0.085 | 0.072 | **0.121** | 0.198 |
| ② Uniform | 0.372 | **0.163** | 0.430 | 0.239 | 0.117 | 0.090 | 0.075 | 0.124 | 0.201 |
| ③ Linear | 0.368 | 0.164 | 0.426 | 0.237 | 0.110 | 0.083 | **0.070** | **0.121** | **0.197** |
| ④ Add | 0.539 | 0.226 | 0.556 | 0.253 | 0.191 | 0.118 | 0.110 | 0.198 | 0.346 |
| ⑤ w/o VAT | 0.373 | 0.172 | 0.437 | 0.245 | 0.123 | 0.098 | 0.111 | 0.186 | 0.218 |
| ⑥ w/o TE | 0.371 | 0.175 | 0.456 | 0.245 | 0.147 | 0.115 | 0.111 | 0.187 | 0.265 |
| ⑦ w/o IN | 0.449 | 0.165 | 0.540 | **0.235** | **0.097** | 0.085 | 0.072 | 0.167 | 0.287 |
| ⑧ with Overlap | 0.371 | 0.164 | 0.427 | 0.238 | 0.113 | **0.082** | 0.073 | 0.122 | 0.199 |
| ⑨ Var-wise | 0.376 | 0.178 | 0.475 | 0.252 | 0.164 | 0.134 | 0.129 | 0.200 | 0.239 |
| ⑩ PatchCat | 0.365 | 0.163 | 0.425 | 0.238 | 0.097 | 0.085 | 0.072 | 0.121 | 0.196 |
| Improve | 2.13% | 7.87% | 10.32% | 5.95% | 32.93% | 38.06% | 45.74% | 39.50% | 21.94% |

the average of three runs with fixed seeds $\{0, 1, 2\}$; the full results and comparison with time-series foundation models are reported in Appendix D.1. As shown in Table 1, our proposed PCMLP achieves state-of-the-art performance across all forecasting error metrics, demonstrating overall leading accuracy. Notably, despite not using complex inter-variable relationship modeling modules like Attention or Transformer, PCMLP still excels in datasets with variable correlation (Liu et al., 2024a), such as Traffic and PEMS07. These experimental results demonstrate the superior capabilities of PatchCat.

Among these baselines, TQNet and CycleNet explicitly model the effects of periodicity, while PatchTST, iTransformer, and TimeXer use attention mechanisms to exploit correlations in the temporal dimension or variable dimension. However, the temporal tokenizers used by these methods only capture local or global semantic information in the time series, limiting their potential for further improvements in predictive performance. Overall, despite PCMLP's simple architecture, it achieves predominant performance. This significant improvement is primarily attributed to our proposed novel temporal tokenizer, PatchCat, which effectively enhances the model's ability to capture local semantics and sequence information.

## 4.3 EXPERIMENTAL STUDIES AND ANALYSIS ABOUT PATCHCAT

We conduct experiments to verify the effectiveness of the dimension-allocation strategy, main operations, hyperparameters, integration ability, and varying lookback lengths of PatchCat.

**Ablation study of dimension allocation strategy.** In addition to the linear strategy described in Section 3.1, we evaluate the performance of the group strategy and the uniform strategy. The group strategy evenly divides all tokens into three groups in chronological order. The output dimension in the projection of the first group is $d$, of the second group is $2d$, and of the third group is $3d$, where the first group is the oldest, and the third group is the closest. The uniform strategy uses one projection for all patches, meaning that each embedding has the same dimension. As shown in Table 2, experimental results show that the group ① and the linear ③ strategies generally outperform the uniform strategy ②. Moreover, the linear strategy achieves top-2 performance on all datasets and achieves optimal performance on average. The increasing strategy (linear and group) improves prediction performance by allocating the embeddings of the most recent patches more representational dimensions, addressing the *recent matters most* phenomenon.

**Ablation study of main operations.** First, we study *the impact of concatenation operation*. We compare the performance of concatenation and addition: the concatenation strategy ⑩ generally outperforms the addition strategy ④. We infer that the key to the performance improvement of concatenation is that it preserves more local details and the order information between patch embeddings. Next, we study *the impact of variable affine transform (VAT)*. Experimental results show that the model with VAT ⑨ outperforms the model without VAT ⑤. We infer that the key

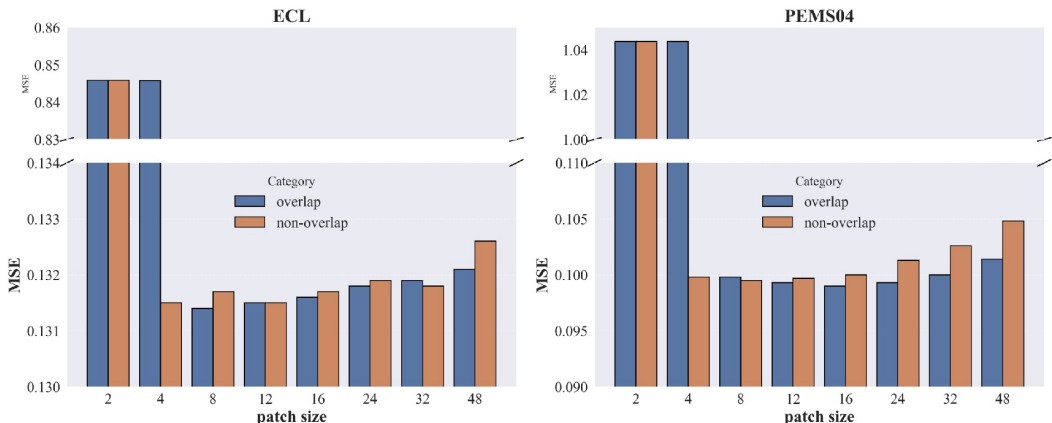

Figure 6: Performance of the PatchCat with varied patch size $S$. Forecasting performance under different patching settings with the lookback length $L = 96$ and prediction length $H = 96$.

to the performance improvement of the VAT module is that it makes patterns in which different variables may have similar histories but different futures distinguishable. We also compare the performance of *with ot without overlaping in patching*. No-overlaping ⑩ outperforms overlaping ⑨ in most cases. Finally, we use PCMLP as a baseline to compare the performance of *PatchCat vs. the variable-wise tokenizer*. Our method ⑩ outperforms the variable-wise tokenizer ⑨ in all cases, with a mean MSE reduction of 21.9%.

**Hyperparameter analysis of PatchCat.** We study the impact of the patch size ($S$) in PatchCat on prediction performance. As shown in Figure 6, experiments on the two datasets show that when $S = 2$, the model's prediction MSE is very large; as $S$ increases, the model's prediction error rapidly decreases and remains stable; and as $S$ continues to increase ($S > 16$), the prediction error begins to increase. It is worth noting that although the optimal $S$ is related to the dataset, to compare fairly with the baseline PatchTST, our experimental results in Table 1 are all based on $S = 16$. We also study whether overlap in patching is necessary. Under the optimal $S$, although the overlapping can slightly reduce the MSE, the effect is not significant and the gap is less than 0.2%.

**Integration analysis of PatchCat.** Besides confirming PatchCat's key role in PCMLP, we further analyze its integration and effectiveness in enhancing the long-term forecasting capabilities of existing models. Specifically, we integrate PatchCat into three powerful time series forecasting models with different network structures to extensively evaluate its impact on the forecasting performance of different kinds of neural networks: iTransformer (Liu et al., 2024a) and PatchTST (Nie et al., 2023) (based on Transformer), MMK (Han et al., 2025) (based on Kolmogorov–Arnold Networks (Liu et al., 2025b)), and DLinear (Zeng et al., 2023) (based on Linear). The results in Table 3 show that integrating the PatchCat consistently improves the original models' performance. This demonstrates the versatility and effectiveness of our proposed PatchCat in improving the predictive capabilities of diverse forecasting methods and its portability across different network architectures. In addition, after replacing the original patch-wise tokenizer of PatchTST with PatchCat, not only achieving significant performance improvements, but also reducing the computational cost to 1/10 of the original. PatchCat achieves an **average performance gain of 5.45%**, indicating that the improvements are consistent and non-trivial.

**Varying lookback length analysis.** We study the effects of different PatchCat settings on prediction performance under varying lookback lengths $L$. As shown in Figure 7, with the lookback length increasing, the prediction MSE under all four settings shows a downward trend, which demonstrates PatchCat's ability to handle different input lengths. More experiments between PatchCat and variable-wise under varying input lengths are reported in Appendix E Table 16.

Table 3: Integration study of the PatchCat. We replace the tokenizer of iTransformer, MMK, DLinear, and PatchTST with our proposed PatchCat.

| Models | iTransformer (2024a) | | | | MMK (2025) | | | | DLinear (2023) | | | | PatchTST (2023) | | | |
|---|---|---|---|---|---|---|---|---|---|---|---|---|---|---|---|---|
| Setup | Original | | +PatchCat | | Original | | +PatchCat | | Original | | +PatchCat | | Original | | +PatchCat | |
| Metrics | MSE | MAE | MSE | MAE | MSE | MAE | MSE | MAE | MSE | MAE | MSE | MAE | MAE | MSE | MAE | MSE |
| ETTh1 96 | 0.386 | 0.405 | **0.380** | **0.398** | 0.374 | 0.397 | **0.372** | **0.395** | 0.386 | 0.400 | **0.379** | **0.391** | 0.414 | 0.419 | **0.382** | **0.399** |
| 192 | 0.441 | 0.436 | **0.432** | **0.429** | 0.419 | 0.429 | **0.419** | **0.425** | 0.437 | 0.432 | **0.430** | **0.421** | 0.460 | 0.445 | **0.435** | **0.429** |
| 336 | 0.487 | 0.458 | **0.472** | **0.446** | 0.461 | 0.450 | **0.456** | **0.444** | 0.481 | 0.459 | **0.474** | **0.448** | 0.501 | 0.466 | **0.475** | **0.448** |
| 720 | 0.503 | 0.491 | **0.484** | **0.475** | 0.474 | 0.467 | **0.465** | **0.462** | 0.519 | 0.516 | **0.489** | **0.479** | 0.500 | 0.488 | **0.476** | **0.471** |
| *Avg* | 0.454 | 0.448 | **0.442** | **0.437** | 0.432 | 0.436 | **0.428** | **0.432** | 0.456 | 0.452 | **0.443** | **0.435** | 0.469 | 0.455 | **0.442** | **0.437** |
| ECL 96 | 0.148 | 0.240 | **0.135** | **0.228** | 0.166 | 0.256 | **0.157** | **0.251** | 0.197 | 0.282 | **0.188** | **0.269** | 0.181 | 0.270 | **0.139** | **0.232** |
| 192 | 0.162 | 0.253 | **0.154** | **0.245** | 0.187 | 0.274 | **0.171** | **0.261** | 0.196 | 0.285 | **0.189** | **0.272** | 0.188 | 0.274 | **0.156** | **0.248** |
| 336 | 0.178 | 0.269 | **0.165** | **0.259** | 0.204 | 0.290 | **0.190** | **0.279** | 0.209 | 0.301 | **0.202** | **0.288** | 0.204 | 0.293 | **0.181** | **0.274** |
| 720 | 0.225 | 0.317 | **0.193** | **0.285** | 0.247 | 0.323 | **0.228** | **0.311** | 0.245 | 0.333 | **0.238** | **0.320** | 0.246 | 0.324 | **0.195** | **0.287** |
| *Avg* | 0.178 | 0.270 | **0.162** | **0.254** | 0.201 | 0.286 | **0.186** | **0.275** | 0.212 | 0.300 | **0.204** | **0.288** | 0.205 | 0.290 | **0.168** | **0.260** |
| Traffic 96 | 0.395 | 0.268 | **0.382** | **0.248** | 0.511 | 0.324 | **0.482** | **0.283** | 0.650 | 0.396 | **0.638** | **0.378** | 0.462 | 0.290 | **0.414** | **0.261** |
| 192 | 0.417 | 0.276 | **0.406** | **0.252** | 0.529 | 0.330 | **0.492** | **0.294** | 0.598 | 0.370 | **0.593** | **0.354** | 0.466 | 0.290 | **0.432** | **0.271** |
| 336 | 0.433 | 0.283 | **0.427** | **0.262** | 0.545 | 0.334 | **0.512** | **0.309** | 0.605 | 0.373 | **0.600** | **0.357** | 0.482 | 0.300 | **0.451** | **0.276** |
| 720 | 0.467 | 0.302 | **0.460** | **0.279** | 0.580 | 0.351 | **0.559** | **0.337** | 0.645 | 0.394 | **0.636** | **0.379** | 0.514 | 0.320 | **0.480** | **0.292** |
| *Avg* | 0.428 | 0.282 | **0.419** | **0.260** | 0.542 | 0.335 | **0.511** | **0.306** | 0.625 | 0.383 | **0.617** | **0.367** | 0.481 | 0.300 | **0.444** | **0.275** |

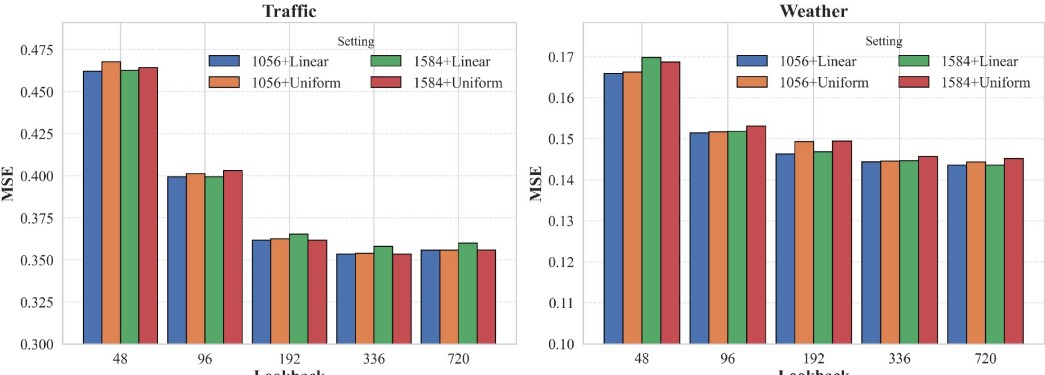

Figure 7: Effects of different PatchCat settings on prediction performance under varying input lengths. Lookback lengths $L = \{48, 96, 192, 336, 720\}$ and prediction length $H = 96$. The bar chart shows the MSE for different combinations of token dimensions $D = \{1056, 1584\}$ and dimension allocation strategies {Linear, Uniform}.

## 4.4 EXPERIMENTAL STUDIES AND ANALYSIS ABOUT PCMLP

**Ablation study of components.** We evaluate the impact of time encoding (TE) and instance normalization (IN) on PCMLP. As shown in Table 2, removing TE ⑥ leads to an increase in the prediction MSE of PCMLP on all datasets, which illustrates the importance of temporal features. In most cases, IN leads to better performance. However, removing IN ⑦ reduces the prediction MSE on Weather and PEMS03.

**Efficiency study.** Although we provide a theoretical analysis of the computational complexity in the Appendix B. We also report the empirical efficiency comparison in Table 4. Compared to Transformer-based models, such as iTransformer, PCMLP has over five times fewer parameters and MACs. Ignoring the influence of backbone, PCiTransformer means replacing the iTransformer's original tokenizer by PatchCat. The reduction in three metrics of computational cost is not only due to the lower MACs of PatchCat itself, but also due to the slight reduction in dimensionality caused by dimension rounding (original tokenizer's $D$=512, PatchCat's

Table 4: Efficiency comparison on the ECL with $L = 96$ and $H = 720$. Training time denotes the average seconds per epoch.

| Model | Parameters | MACs | Training times(s) |
|---|---|---|---|
| Informer | 12.53M | 3.97G | 70.1 |
| Autoformer | 12.22M | 4.41G | 107.7 |
| DLinear | 139.6K | 44.91M | 18.1 |
| CycleNet | 472.9K | 134.84M | 30.8 |
| iTransformer | 6.408M | 2.056G | 35.1 |
| PatchTST | 10.74M | 25.87G | 129.5 |
| PCiTransformer | 6.167M | 1.980G | 34.7 |
| PCMLP | 938.37K | 300.99M | 25.7 |

$D$=504). Although PCMLP is slower than DLinear, considering the significant improvement in prediction accuracy brought, PCMLP achieves the best balance between performance and efficiency.

### 4.5 KEYS TO PATCHCAT'S PERFORMANCE IMPROVEMENT

We conduct two groups of experiments to verify that PatchCat mitigates the two major performance limitations of variable-wise (var-wise) tokenizers. For a fair comparison, all experiments use the MLP as a backbone.

**Modeling temporal order**   We first examine whether PatchCat is better at modeling temporal order information. Each tokenizer is evaluated under three input settings: (1) the original input (Ori), (2) a randomly shuffled input (Shuffle), and (3) an input where the first and second halves of the sequence are exchanged (Half-EX). As shown in Table 5, PatchCat suffers a substantially larger performance drop when the temporal order is shuffled. This indicates that PatchCat captures temporal dependencies more faithfully than the var-wise tokenizer, whose performance remains relatively unaffected by sequence disordering.

Table 5: Comparison of capabilities in modeling temporal order information.

| Setting | ECL | | | PEMS08 | | |
|---|---|---|---|---|---|---|
| | Ori | Shuffle | Half-EX | Ori | Shuffle | Half-EX |
| PatchCat | 0.132 | 1.003 | 0.256 | 0.069 | 1.495 | 1.610 |
| Var-wise Tokenizer | 0.152 | 0.950 | 0.204 | 0.086 | 0.960 | 0.256 |

**Sensitivity to Temporal Mutations**   Next, we evaluate the tokenizer's ability to detect temporal mutations. We define a *temporal mutation* as a local trend reversal occurring within the latest $K$ steps of the input data. Table 6 reports the performance of PatchCat and the var-wise tokenizer on samples containing temporal mutations for $K = \{3, 6, 12\}$. PatchCat responds more effectively to such abrupt local changes due to its linearly increasing dimension allocation, which assigns more embedding capacity to recent patches. Across mutation-containing samples, PatchCat achieves roughly a 15% reduction in MSE compared with the var-wise tokenizer. Visualizations of representative cases are provided in Appendix F.

Table 6: Comparison of sensitivity to temporal mutations.

| Setting | ECL | | | PEMS08 | | |
|---|---|---|---|---|---|---|
| | K=3 | K=6 | K=12 | K=3 | K=6 | K=12 |
| Var-wise Tokenizer | 0.153 | 0.154 | 0.151 | 0.089 | 0.087 | 0.086 |
| PatchCat (Uniform) | 0.147 | 0.148 | 0.145 | 0.080 | 0.078 | 0.077 |
| PatchCat (Linear) | 0.133 | 0.134 | 0.131 | 0.074 | 0.072 | 0.071 |
| Improve | 13.36% | 13.24% | 13.12% | 17.42% | 17.23% | 16.90% |

## 5   CONCLUSION

We revisit temporal tokenizers for time series forecasting and evaluate point-wise, patch-wise, and variable-wise strategies within a unified forecasting framework. Our analysis reveals a fundamental trade-off: patch-wise designs deliver higher accuracy by leveraging local semantics, whereas variable-wise designs achieve superior efficiency via compact representations. Crucially, these strengths are not mutually exclusive, motivating new designs that reconcile both. To this end, we proposed PatchCat, a **patch-then-concat** tokenizer that embeds local temporal relationships into a single token while retaining computational efficiency. Augmented with dimension allocation and variable-aware affine transformation, PatchCat enhances representational capacity and consistently outperforms strong baselines across 13 real-world datasets with a simple MLP backbone. Our work clarifies the role of tokenization in TSF, paving the way for future advances.

## 6 REPRODUCIBILITY STATEMENT

We have provided all information necessary to reproduce the main experimental results of this work, sufficient to support its central claims and conclusions. In detail, the pseudocode of core algorithm is provided in Appendix B, the experimental settings are described in Section 4.1, and the full source code is included in the supplementary material.

## 7 ETHICS STATEMENT

This research relies solely on publicly available datasets and does not involve the collection of sensitive information or human subjects. We have taken care to respect the licenses and usage terms of all datasets and code bases. While our method may be applied across a range of domains, we recognize the potential for misuse in generating harmful or biased outputs. To mitigate this risk, we provide thorough documentation and encourage responsible use of our approach within ethical and socially beneficial contexts.

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

## A  COMPARISON OF TEMPORAL TOKENIZERS

We study the impact of different temporal tokenizers on time series forecasting. Although each kind of tokenizer is used by many methods, these methods are based on different backbone networks, making it difficult to fairly reflect the impact of the tokenizer module on the final prediction performance. To this end, we use a general forecasting pipeline for evaluation. As shown in Figure 8, this pipeline is modified from PatchTST (Nie et al., 2023), consisting of a temporal tokenizer, stacked multi-layer Transformer layers, an output projection, and instance normalization. According to the analysis in Section 1, point-wise can be regarded as a special case of patch-wise tokenizer with patch size = 1, and variable-wise can be regarded as a special case of patch-wise tokenizer with patch size equal to input length. Therefore, we use PatchTST's tokenizer as a unified implementation of point/patch/variable-wise tokenizers. By adjusting the patch size, we obtain the prediction accuracy and computational cost of the three types of tokenizers under the same backbone network, achieving a fair comparison. For point/patch-wise tokenizers, the transformers capture the temporal dependency of tokens. For variable tokenizer and PatchCat, the transformers capture the autocorrelation of tokens.

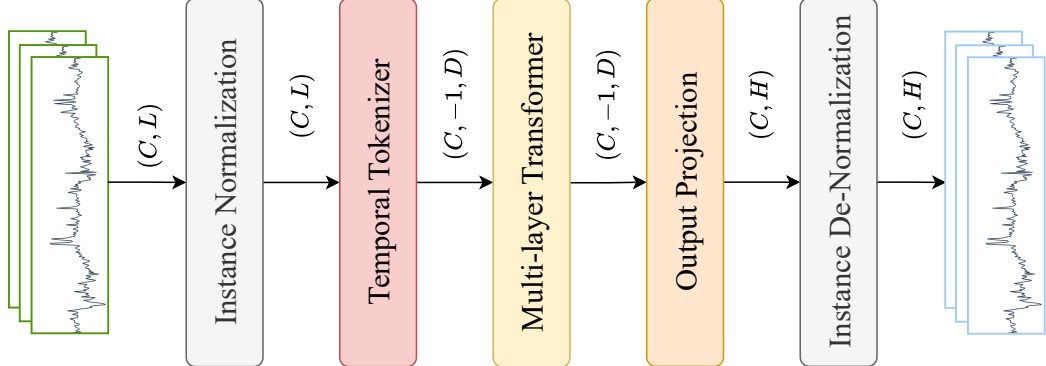

Figure 8: General pipeline for four kinds of temporal tokenizer. This pipeline is modified from PatchTST.

We search for three key hyperparameters that influence performance: the token dimension (d_model), the number of transformer layers (n_layer), and the learning rate. Their search space is shown in Table 7. This setting enhances the stability of the experimental results in Table 8. On the Traffic dataset, the patch-wise setting achieves the better MSE and MAE with $S = 16$ than the point-wise and variable-wise tokenizers. However, its MACs are approximately four times that of the variable-wise tokenizer. Similar results are obtained on the Weather dataset. Moreover, our proposed PatchCat achieves the best results in terms of computational overhead and prediction accuracy.

Table 7: Hyperparameter search space in temporal tokenizer comparative experiments.

| Hyperparameter | Search space |
|---|---|
| d_model | {128, 256, 512} |
| n_layer | {2, 3, 4} |
| learning rate | {0.001, 0.0005, 0.0001} |

## B  DETAILS ABOUT PATCHCAT

**Pseudocode of PatchCat.**  As shown in Figure 4, PatchCat consists of four sequential steps: patching, projection, concatenation, and transformation. The pseudocode of PatchCat is shown in Alg 1. First, PatchCat segments input time series data $X$ into patches $X'$ with patch size $S$. There is no overlapping during patching. Then, the learnable linear layers are applied to project $X'$ into embeddings with different output dimensions. We use a linearly increasing dimension allocation strategy,

Table 8: Comparison of three kinds of temporal tokenizer. The lookback length $L$ is 96 and forecasting length $H$ is 96.

| Setting | $S$ | Traffic | | | Weather | | |
|---|---|---|---|---|---|---|---|
| | | MSE | MAE | MACs | MSE | MAE | MACs |
| point-wise | 1 | 0.4377 | 0.2890 | 558.119G | 0.1732 | 0.2131 | 13.597G |
| patch-wise | 4 | 0.4188 | 0.2752 | 143.878G | 0.1730 | 0.2118 | 3.505G |
| | 8 | 0.4139 | 0.2617 | 74.840G | 0.1727 | 0.2107 | 1.823G |
| | 12 | 0.4143 | 0.2663 | 51.828G | 0.1753 | 0.2124 | 1.263G |
| | 16 | 0.4126 | 0.2643 | 40.323G | 0.1737 | 0.2106 | 982.346M |
| | 24 | 0.4178 | 0.2677 | 28.820G | 0.1735 | 0.2108 | 702.106M |
| | 32 | 0.4180 | 0.2684 | 23.070G | 0.1755 | 0.2109 | 562.029M |
| variable-wise | 96 | 0.4252 | 0.2708 | 6.591G | 0.1778 | 0.2125 | 142.391M |
| PatchCat | 12 | **0.4000** | **0.2579** | **5.520G** | **0.1569** | **0.1974** | **129.506M** |

which means that we assign larger dimensions to the nearer patch embedding. Subsequently, we concatenate the embeddings in chronological order to obtain one token $\mathbf{T}$ with dimension $D$. Finally, PatchCat uses an affine transformation on tokens along the variable dimension to enhance inter-variable token distinguishability.

---

**Algorithm 1** The pipeline of PatchCat

---

**Require:** Input time series $X \in \mathbb{R}^{B \times C \times L}$, patch size $S$, dimension of the first embedding $d$.
**Ensure:** Output tokens $\mathbf{T} \in \mathbb{R}^{B \times C \times D}$
1: Patchify input time series: $X \leftarrow \text{patch}(X, \text{size} = S)$ ▷ Shape: $(B, C, \frac{L}{S}, S)$
2: $\mathbf{T} \leftarrow \emptyset$
3: **for** $i$ in range$\left(\frac{L}{S}\right)$ **do** ▷ Process each dimension group
4:     Project: $\mathbf{T}_i \leftarrow \text{Linear}_i(X[:, :, i, :])$ ▷ Shape: $(B, C, i \times d)$
5:     Append: $\mathbf{T} \leftarrow \mathbf{T} \cup \{\mathbf{T}_i\}$
6: **end for**
7: Concatenate: $\mathbf{T} \leftarrow \text{concat}(\mathbf{T}, \dim = -1)$ ▷ Shape: $(B, C, D)$
8: Transform: $\mathbf{T} \leftarrow \alpha \mathbf{T} + \beta$ ▷ Shape: $(B, C, D)$
9: **return** $\mathbf{T}$

---

**Complexity analysis of PatchCat.** We use MACs to measure the computational complexity of PatchCat. PatchCat's computational complexity consists of two parts: MACs of projection $\text{MACs}_{\text{proj}}$ and MACs of transform $\text{MACs}_{\text{trn}}$:

$$\text{MACs}_{\text{proj}} = S \times d + S \times 2d + \cdots + S \times \frac{d \times L}{S} = S \times D,$$

$$\text{MACs}_{\text{trn}} = d + 2d + \cdots + \frac{d \times L}{S} = D, \tag{7}$$

$$\text{MACs}_{\text{PatchCat}} = \text{MACs}_{\text{proj}} + \text{MACs}_{\text{trn}} = (S + 1) \times D$$

where $L$ is the input time series length, $S$ is the patch size, $d$ is the dimension of the first embedding, and $D$ is the dimension of the token. Specifically, $D = d + 2d + \cdots + \frac{d \times L}{S}$. We compare the four temporal tokenizers in Table 9, since $L$ is much larger than $S$, our PatchCat has the smallest computational cost when $D$ is the same.

Table 9: The MACs of four temporal tokenizers.

| PatchCat | Point-wise | Patch-wise | Variable-wise |
|---|---|---|---|
| $(S + 1) \times D$ | $L \times D$ | $L \times D$ | $L \times D$ |

## C  Dataset Descriptions

As shown in Table C, we use 11 real-world time series forecasting datasets, including ETT series (Zhou et al., 2021) (ETTh1, ETTh2, ETTm1, and ETTm2), Traffic, Weather (Wu et al., 2021), Electricity (Lai et al., 2018) (abbreviated as ECL), Solar-Energy (Solar) (Lai et al., 2018), Beijing Air Quality (BJAQ), and PEMS series (PEMS03, PEMS04, PEMS07, and PEMS08) (Guo et al., 2021). Following previous work (Liu et al., 2024a), we split ETT series and PEMS series with 6:2:2, and others with 7:1:2.

Table 10: Datasets summary.

| Dataset | Channels | Length | Interval | Domain |
|---------|----------|--------|----------|--------|
| ETTh1 | 7 | 14,400 | 1 hour | Electricity |
| ETTh2 | 7 | 14,400 | 1 hour | Electricity |
| ETTm1 | 7 | 57,600 | 15 mins | Electricity |
| ETTm2 | 7 | 57,600 | 15 mins | Electricity |
| ECL | 321 | 26,304 | 1 hour | Electricity |
| Traffic | 862 | 17,544 | 1 hour | Transportation |
| Weather | 21 | 52,696 | 10 mins | Weather |
| PEMS03 | 358 | 26,208 | 5 mins | Transportation |
| PEMS04 | 307 | 16,992 | 5 mins | Transportation |
| PEMS07 | 883 | 28,224 | 5 mins | Transportation |
| PEMS08 | 170 | 17,856 | 5 mins | Transportation |
| Solar | 7 | 52,560 | 10 mins | Nature |
| BJAQ | 7 | 36,000 | 1 hour | Environment |

## D  More Experimental Results

### D.1  Full Results of Overall Performance

We compare the forecasting errors of PCMLP with eight baseline models on 11 real-world datasets, including TQNet (Lin et al., 2025), TimeXer (Wang et al., 2024), CycleNet (Lin et al., 2024), iTransformer (Liu et al., 2024a), MSGNet (Cai et al., 2024), TimesNet (Wu et al., 2023), PatchTST (Nie et al., 2023), and DLinear (Zeng et al., 2023). The results of our method are the average of three runs with fixed seeds $\{0, 1, 2\}$. We report the standard deviation of three runnings on all datasets in Table 11, and the full results in Table 13. The hyperparameter search space is shown in Table 12.

As shown in Table 13, our proposed PCMLP achieves state-of-the-art performance across all forecasting error metrics, demonstrating overall leading accuracy. Notably, despite not using complex inter-variable relationship modeling modules like Attention or Transformer, PCMLP still excels in datasets with a large number of variables, such as Traffic and PEMS07 (more than 800 variables). These experimental results demonstrate the superior capabilities of PatchCat. Among these baselines, TQNet and CycleNet explicitly model the effects of periodicity, while PatchTST, iTransformer, and TimeXer use attention mechanisms to exploit correlations in the temporal dimension or variable dimension. However, the temporal tokenizers used by these methods only capture local or global semantic information in the time series, limiting their potential for further improvements in predictive performance. Overall, despite PCMLP's simple architecture, it achieves predominant performance. This significant improvement is primarily attributed to our proposed novel temporal tokenizer, PatchCat, which effectively enhances the model's ability to capture local semantics and sequence information.

We further compare PCMLP with time-series foundation models (TFMs) (Das et al., 2024; Ansari et al., 2024; Goswami et al., 2024; Żukowska et al., 2024; Shchur et al., 2025) following the evaluation protocol of TIMER-XL (Liu et al., 2025a). Under the *96-pred-96* setting across five public datasets, PCMLP achieves the best performance in 9 out of 10 cases as shown in Table 14. Moreover, it is substantially more parameter-efficient: on ETTh1, PCMLP uses only 2.8% of TIMER-XL's parameters (2.4 MB vs. 84 MB) while reducing MAE by 2.88%.

Table 11: Full standard deviations of long-term multivariate time series forecasting results with prediction lengths $H \in \{12, 24, 48, 96\}$ for PEMS and $H \in \{96, 192, 336, 720\}$ for others. The lookback length $L = 96$ for all experiments. Results are calculated over three runs with fixed seeds $\{0, 1, 2\}$.

| H | ETTh1 | | ETTh2 | | ETTm1 | | ETTm2 | |
|---|---|---|---|---|---|---|---|---|
| | MSE | MAE | MSE | MAE | MSE | MAE | MSE | MAE |
| 96 | 0.0021 | 0.0018 | 0.0008 | 0.0006 | 0.0009 | 0.0011 | 0.0019 | 0.0021 |
| 192 | 0.0049 | 0.0007 | 0.0008 | 0.0009 | 0.0011 | 0.0006 | 0.0001 | 0.0002 |
| 336 | 0.0036 | 0.0019 | 0.0027 | 0.0009 | 0.0020 | 0.0014 | 0.0005 | 0.0004 |
| 720 | 0.0236 | 0.0129 | 0.0012 | 0.0015 | 0.0014 | 0.0019 | 0.0012 | 0.0015 |

| H | PEMS03 | | PEMS04 | | PEMS07 | | PEMS08 | |
|---|---|---|---|---|---|---|---|---|
| | MSE | MAE | MSE | MAE | MSE | MAE | MSE | MAE |
| 12 | 0.0002 | 0.0002 | 0.0001 | 0.0002 | 0.0001 | 0.0001 | 0.0001 | 0.0002 |
| 24 | 0.0003 | 0.0003 | 0.0001 | 0.0002 | 0.0001 | 0.0002 | 0.0005 | 0.0003 |
| 48 | 0.0007 | 0.0008 | 0.0002 | 0.0002 | 0.0002 | 0.0009 | 0.0005 | 0.0010 |
| 96 | 0.0006 | 0.0007 | 0.0007 | 0.0008 | 0.0003 | 0.0003 | 0.0027 | 0.0026 |

| H | ECL | | Traffic | | Weather | | | |
|---|---|---|---|---|---|---|---|---|
| | MSE | MAE | MSE | MAE | MSE | MAE | | |
| 96 | 0.0001 | 0.0001 | 0.0007 | 0.0001 | 0.0011 | 0.0013 | | |
| 192 | 0.0001 | 0.0001 | 0.0004 | 0.0004 | 0.0003 | 0.0003 | | |
| 336 | 0.0001 | 0.0001 | 0.0017 | 0.0000 | 0.0016 | 0.0014 | | |
| 720 | 0.0009 | 0.0005 | 0.0019 | 0.0003 | 0.0009 | 0.0006 | | |

Table 12: Hyperparameter search space of PCMLP.

| Hyperparameter | Search space |
|---|---|
| D | {264, 528, 1056} |
| n_layer | {1, 2} |
| dropout rate | {0.1, 0.2} |
| learning rate | {0.01, 0.005, 0.001, 0.0005} |

## D.2 FULL RESULTS OF ABLATION STUDY

We report the full ablation study in Table 15. We study the impact of each operation in PatchCat on the prediction performance. Firstly, we study *the impact of dimension allocation strategy*. In addition to the linear strategy described in Section 3.1, we evaluate the performance of the group strategy and the uniform strategy. The group strategy evenly divides all tokens into three groups in chronological order. The output dimension in the projection of the first group is $d$, of the second group is $2d$, and of the third group is $3d$, where the first group is the oldest, and the third group is the closest. The uniform strategy uses one projection for all patches, meaning that each embedding has the same dimension. As shown in Table 2, experimental results show that the group ① and the linear ③ strategies generally outperform the uniform strategy ②. Moreover, the linear strategy achieves top-2 performance on all datasets and achieves optimal performance on average. The increasing strategy (linear and group) improves prediction performance by allocating the embeddings of the most recent patches more representational dimensions, addressing the *recent matters most* phenomenon. Then, we study *the impact of the concatenation operation*. We compare the performance of concatenation and addition: the concatenation strategy ⑨ generally outperforms the addition strategy ④. We infer that the key to the performance improvement of concatenation is that it preserves more local details and the order information between patch embeddings. Next, we study *the impact of variable affine transform (VAT)*. Experimental results show that the model with VAT ⑨ outperforms the model without VAT ⑤. We infer that the key to the performance improvement of the VAT module is that it makes patterns in which different variables may have similar histories but different futures distinguishable. Finally, we use PCMLP as a baseline to compare the

Table 13: Full results of long-term multivariate time series forecasting results with prediction lengths $H \in \{12, 24, 48, 96\}$ for PEMS and $H \in \{96, 192, 336, 720\}$ for others. The lookback length $L = 96$ for all experiments. Results are averaged over three runs with fixed seeds $\{0, 1, 2\}$. The best results are in **red**.

| Model | | PCMLP (Ours) | | TQNet (2025) | | MMK (2025)) | | TimeXer (2024) | | CycleNet (2024) | | iTransformer (2024a) | | MSGNet (2024) | | TimesNet (2023) | | PatchTST (2023) | | DLinear (2023) | |
|---|---|---|---|---|---|---|---|---|---|---|---|---|---|---|---|---|---|---|---|---|---|
| Metric | | MSE | MAE | MSE | MAE | MSE | MAE | MSE | MAE | MSE | MAE | MSE | MAE | MSE | MAE | MSE | MAE | MSE | MAE | MSE | MAE |
| ETTh1 | 96 | **0.370** | **0.391** | 0.371 | 0.393 | 0.374 | 0.397 | 0.382 | 0.403 | 0.375 | 0.395 | 0.386 | 0.405 | 0.390 | 0.411 | 0.384 | 0.402 | 0.414 | 0.419 | 0.386 | 0.400 |
| | 192 | 0.431 | **0.424** | **0.428** | 0.426 | 0.419 | 0.429 | 0.429 | 0.435 | 0.436 | 0.428 | 0.441 | 0.436 | 0.443 | 0.442 | 0.436 | 0.429 | 0.460 | 0.445 | 0.437 | 0.432 |
| | 336 | 0.482 | **0.446** | 0.476 | 0.446 | 0.461 | 0.450 | **0.468** | 0.448 | 0.496 | 0.455 | 0.487 | 0.458 | 0.482 | 0.469 | 0.491 | 0.469 | 0.501 | 0.466 | 0.481 | 0.459 |
| | 720 | 0.503 | 0.474 | 0.487 | 0.470 | 0.474 | 0.467 | **0.469** | **0.461** | 0.520 | 0.484 | 0.503 | 0.491 | 0.496 | 0.488 | 0.521 | 0.500 | 0.500 | 0.488 | 0.519 | 0.516 |
| | Avg | 0.447 | **0.434** | 0.441 | 0.434 | 0.432 | 0.436 | **0.437** | 0.437 | 0.457 | 0.441 | 0.454 | 0.448 | 0.453 | 0.453 | 0.458 | 0.450 | 0.469 | 0.455 | 0.456 | 0.452 |
| ETTh2 | 96 | 0.291 | 0.340 | 0.295 | 0.343 | 0.301 | 0.353 | **0.286** | **0.338** | 0.298 | 0.344 | 0.297 | 0.349 | 0.329 | 0.371 | 0.340 | 0.374 | 0.302 | 0.348 | 0.333 | 0.387 |
| | 192 | 0.368 | 0.391 | 0.367 | 0.393 | 0.379 | 0.405 | **0.363** | **0.389** | 0.372 | 0.396 | 0.380 | 0.400 | 0.402 | 0.414 | 0.388 | 0.400 | 0.388 | 0.400 | 0.477 | 0.476 |
| | 336 | **0.413** | 0.428 | 0.417 | 0.427 | 0.432 | 0.446 | 0.414 | **0.423** | 0.431 | 0.439 | 0.428 | 0.432 | 0.440 | 0.445 | 0.452 | 0.452 | 0.426 | 0.433 | 0.594 | 0.541 |
| | 720 | 0.423 | 0.442 | 0.433 | 0.446 | 0.446 | 0.463 | **0.408** | **0.432** | 0.450 | 0.458 | 0.427 | 0.445 | 0.480 | 0.477 | 0.462 | 0.468 | 0.431 | 0.446 | 0.831 | 0.657 |
| | Avg | 0.374 | 0.400 | 0.378 | 0.402 | 0.390 | 0.417 | **0.368** | **0.396** | 0.388 | 0.409 | 0.383 | 0.407 | 0.413 | 0.427 | 0.414 | 0.427 | 0.387 | 0.407 | 0.559 | 0.515 |
| ETTm1 | 96 | **0.305** | **0.344** | 0.311 | 0.353 | 0.320 | 0.358 | 0.318 | 0.356 | 0.319 | 0.360 | 0.334 | 0.368 | 0.319 | 0.366 | 0.338 | 0.375 | 0.329 | 0.367 | 0.345 | 0.372 |
| | 192 | 0.358 | **0.376** | **0.356** | **0.378** | 0.364 | 0.383 | 0.362 | 0.383 | 0.360 | 0.381 | 0.377 | 0.391 | 0.377 | 0.397 | 0.374 | 0.387 | 0.367 | 0.385 | 0.380 | 0.389 |
| | 336 | **0.386** | **0.395** | 0.390 | 0.401 | 0.395 | 0.405 | 0.395 | 0.407 | 0.389 | 0.403 | 0.426 | 0.420 | 0.417 | 0.422 | 0.410 | 0.411 | 0.399 | 0.410 | 0.413 | 0.413 |
| | 720 | **0.444** | **0.436** | 0.452 | 0.440 | 0.457 | 0.440 | 0.452 | 0.441 | 0.447 | 0.441 | 0.491 | 0.459 | 0.487 | 0.463 | 0.478 | 0.450 | 0.454 | 0.439 | 0.474 | 0.453 |
| | Avg | **0.373** | **0.388** | 0.377 | 0.393 | 0.384 | 0.397 | 0.382 | 0.397 | 0.379 | 0.396 | 0.407 | 0.410 | 0.400 | 0.412 | 0.400 | 0.406 | 0.387 | 0.400 | 0.403 | 0.407 |
| ETTm2 | 96 | 0.168 | 0.248 | 0.173 | 0.256 | 0.176 | 0.261 | 0.171 | 0.256 | **0.163** | **0.246** | 0.180 | 0.264 | 0.182 | 0.266 | 0.187 | 0.267 | 0.175 | 0.259 | 0.193 | 0.292 |
| | 192 | **0.229** | **0.289** | 0.238 | 0.298 | 0.240 | 0.302 | 0.237 | 0.299 | **0.229** | 0.290 | 0.250 | 0.309 | 0.248 | 0.306 | 0.249 | 0.309 | 0.241 | 0.302 | 0.284 | 0.362 |
| | 336 | 0.287 | 0.328 | 0.301 | 0.340 | 0.299 | 0.342 | 0.296 | 0.338 | **0.284** | **0.327** | 0.311 | 0.348 | 0.312 | 0.346 | 0.321 | 0.351 | 0.305 | 0.343 | 0.369 | 0.427 |
| | 720 | **0.388** | **0.388** | 0.397 | 0.396 | 0.397 | 0.401 | 0.392 | 0.394 | 0.389 | 0.391 | 0.412 | 0.407 | 0.414 | 0.404 | 0.408 | 0.403 | 0.402 | 0.400 | 0.554 | 0.522 |
| | Avg | 0.268 | **0.313** | 0.277 | 0.323 | 0.278 | 0.327 | 0.274 | 0.322 | **0.266** | 0.314 | 0.288 | 0.332 | 0.289 | 0.330 | 0.291 | 0.333 | 0.281 | 0.326 | 0.350 | 0.401 |
| ECL | 96 | **0.132** | **0.223** | 0.134 | 0.229 | 0.166 | 0.256 | 0.140 | 0.242 | 0.136 | 0.229 | 0.148 | 0.240 | 0.165 | 0.274 | 0.168 | 0.272 | 0.181 | 0.270 | 0.197 | 0.282 |
| | 192 | **0.150** | **0.239** | 0.154 | 0.247 | 0.187 | 0.274 | 0.157 | 0.256 | 0.152 | 0.244 | 0.162 | 0.253 | 0.185 | 0.292 | 0.184 | 0.289 | 0.188 | 0.274 | 0.196 | 0.285 |
| | 336 | **0.167** | **0.257** | 0.169 | 0.264 | 0.204 | 0.290 | 0.176 | 0.275 | 0.170 | 0.264 | 0.178 | 0.269 | 0.197 | 0.304 | 0.198 | 0.300 | 0.204 | 0.293 | 0.209 | 0.301 |
| | 720 | 0.204 | **0.291** | **0.201** | 0.294 | 0.247 | 0.323 | 0.211 | 0.306 | 0.212 | 0.299 | 0.225 | 0.317 | 0.231 | 0.332 | 0.220 | 0.320 | 0.246 | 0.324 | 0.245 | 0.333 |
| | Avg | **0.163** | **0.252** | 0.164 | 0.259 | 0.201 | 0.286 | 0.171 | 0.270 | 0.168 | 0.259 | 0.178 | 0.270 | 0.194 | 0.301 | 0.193 | 0.295 | 0.205 | 0.290 | 0.212 | 0.300 |
| Traffic | 96 | 0.396 | **0.244** | 0.413 | 0.261 | 0.511 | 0.324 | 0.428 | 0.271 | 0.458 | 0.296 | **0.395** | 0.268 | 0.608 | 0.349 | 0.593 | 0.321 | 0.462 | 0.290 | 0.650 | 0.396 |
| | 192 | **0.411** | **0.253** | 0.432 | 0.271 | 0.529 | 0.330 | 0.448 | 0.282 | 0.457 | 0.294 | 0.417 | 0.276 | 0.634 | 0.371 | 0.617 | 0.336 | 0.466 | 0.290 | 0.598 | 0.37 |
| | 336 | **0.428** | **0.259** | 0.450 | 0.277 | 0.545 | 0.334 | 0.473 | 0.289 | 0.470 | 0.299 | 0.433 | 0.283 | 0.669 | 0.388 | 0.629 | 0.336 | 0.482 | 0.300 | 0.605 | 0.373 |
| | 720 | **0.466** | **0.279** | 0.486 | 0.295 | 0.580 | 0.351 | 0.516 | 0.307 | 0.502 | 0.314 | 0.467 | 0.302 | 0.729 | 0.420 | 0.640 | 0.350 | 0.514 | 0.320 | 0.645 | 0.394 |
| | Avg | **0.425** | **0.259** | 0.445 | 0.276 | 0.542 | 0.335 | 0.466 | 0.287 | 0.472 | 0.301 | 0.428 | 0.282 | 0.660 | 0.382 | 0.620 | 0.336 | 0.481 | 0.300 | 0.625 | 0.383 |
| Weather | 96 | **0.151** | **0.191** | 0.157 | 0.200 | 0.171 | 0.221 | 0.157 | 0.205 | 0.158 | 0.203 | 0.174 | 0.214 | 0.163 | 0.212 | 0.172 | 0.220 | 0.177 | 0.210 | 0.196 | 0.255 |
| | 192 | **0.200** | **0.238** | 0.206 | 0.245 | 0.220 | 0.263 | 0.204 | 0.247 | 0.207 | 0.247 | 0.221 | 0.254 | 0.211 | 0.254 | 0.219 | 0.261 | 0.225 | 0.250 | 0.237 | 0.296 |
| | 336 | **0.258** | **0.282** | 0.262 | 0.287 | 0.277 | 0.302 | 0.261 | 0.290 | 0.262 | 0.289 | 0.278 | 0.296 | 0.273 | 0.299 | 0.280 | 0.306 | 0.278 | 0.290 | 0.283 | 0.335 |
| | 720 | 0.342 | **0.339** | 0.344 | 0.342 | 0.360 | 0.354 | **0.340** | 0.341 | 0.344 | 0.344 | 0.358 | 0.349 | 0.351 | 0.348 | 0.365 | 0.359 | 0.354 | 0.340 | 0.345 | 0.381 |
| | Avg | **0.238** | **0.263** | 0.242 | 0.269 | 0.257 | 0.285 | 0.241 | 0.271 | 0.243 | 0.271 | 0.258 | 0.278 | 0.249 | 0.278 | 0.259 | 0.287 | 0.259 | 0.273 | 0.265 | 0.317 |
| PEMS03 | 12 | 0.062 | 0.162 | **0.060** | **0.161** | 0.087 | 0.195 | 0.070 | 0.173 | 0.066 | 0.172 | 0.071 | 0.174 | 0.078 | 0.187 | 0.085 | 0.192 | 0.099 | 0.216 | 0.122 | 0.243 |
| | 24 | 0.079 | 0.180 | **0.077** | **0.182** | 0.146 | 0.253 | 0.092 | 0.194 | 0.089 | 0.201 | 0.093 | 0.201 | 0.108 | 0.218 | 0.118 | 0.223 | 0.142 | 0.259 | 0.201 | 0.317 |
| | 48 | 0.109 | **0.205** | **0.104** | 0.215 | 0.292 | 0.368 | 0.129 | 0.229 | 0.136 | 0.247 | 0.125 | 0.236 | 0.178 | 0.272 | 0.155 | 0.260 | 0.211 | 0.319 | 0.333 | 0.425 |
| | 96 | **0.136** | **0.226** | 0.148 | 0.253 | 0.504 | 0.502 | 0.157 | 0.261 | 0.182 | 0.282 | 0.164 | 0.275 | 0.238 | 0.328 | 0.228 | 0.317 | 0.269 | 0.370 | 0.457 | 0.515 |
| | Avg | **0.097** | **0.193** | 0.097 | 0.203 | 0.257 | 0.329 | 0.112 | 0.214 | 0.118 | 0.226 | 0.113 | 0.222 | 0.150 | 0.251 | 0.147 | 0.248 | 0.180 | 0.291 | 0.278 | 0.375 |
| PEMS04 | 12 | 0.068 | 0.167 | **0.067** | **0.166** | 0.103 | 0.212 | 0.074 | 0.178 | 0.078 | 0.186 | 0.078 | 0.183 | 0.086 | 0.199 | 0.087 | 0.195 | 0.105 | 0.224 | 0.148 | 0.272 |
| | 24 | 0.078 | **0.178** | **0.077** | 0.181 | 0.169 | 0.276 | 0.087 | 0.195 | 0.099 | 0.212 | 0.095 | 0.205 | 0.101 | 0.218 | 0.103 | 0.215 | 0.153 | 0.275 | 0.224 | 0.340 |
| | 48 | **0.091** | **0.192** | 0.097 | 0.206 | 0.323 | 0.391 | 0.110 | 0.214 | 0.133 | 0.248 | 0.120 | 0.233 | 0.127 | 0.247 | 0.136 | 0.250 | 0.229 | 0.339 | 0.355 | 0.437 |
| | 96 | **0.102** | **0.203** | 0.123 | 0.233 | 0.568 | 0.539 | 0.148 | 0.251 | 0.167 | 0.281 | 0.150 | 0.262 | 0.174 | 0.292 | 0.190 | 0.303 | 0.291 | 0.389 | 0.452 | 0.504 |
| | Avg | **0.085** | **0.185** | 0.091 | 0.197 | 0.291 | 0.354 | 0.105 | 0.209 | 0.119 | 0.232 | 0.111 | 0.221 | 0.122 | 0.239 | 0.129 | 0.241 | 0.195 | 0.307 | 0.295 | 0.388 |
| PEMS07 | 12 | 0.052 | **0.142** | **0.051** | 0.143 | 0.081 | 0.187 | 0.057 | 0.152 | 0.062 | 0.162 | 0.067 | 0.165 | 0.079 | 0.182 | 0.082 | 0.181 | 0.095 | 0.207 | 0.115 | 0.242 |
| | 24 | **0.062** | **0.154** | 0.063 | 0.159 | 0.142 | 0.249 | 0.079 | 0.179 | 0.086 | 0.192 | 0.088 | 0.190 | 0.099 | 0.206 | 0.101 | 0.204 | 0.150 | 0.262 | 0.210 | 0.329 |
| | 48 | **0.077** | **0.169** | 0.081 | 0.179 | 0.289 | 0.363 | 0.099 | 0.191 | 0.128 | 0.234 | 0.110 | 0.215 | 0.133 | 0.239 | 0.134 | 0.238 | 0.253 | 0.340 | 0.398 | 0.458 |
| | 96 | **0.097** | **0.184** | 0.103 | 0.203 | 0.547 | 0.512 | 0.107 | 0.205 | 0.176 | 0.268 | 0.139 | 0.245 | 0.179 | 0.279 | 0.181 | 0.279 | 0.346 | 0.404 | 0.594 | 0.553 |
| | Avg | **0.072** | **0.162** | 0.075 | 0.171 | 0.265 | 0.328 | 0.085 | 0.182 | 0.113 | 0.214 | 0.101 | 0.204 | 0.122 | 0.227 | 0.125 | 0.226 | 0.211 | 0.303 | 0.329 | 0.396 |
| PEMS08 | 12 | **0.069** | **0.162** | 0.071 | 0.170 | 0.099 | 0.204 | 0.075 | 0.176 | 0.082 | 0.185 | 0.079 | 0.182 | 0.105 | 0.211 | 0.112 | 0.212 | 0.168 | 0.232 | 0.154 | 0.276 |
| | 24 | **0.089** | **0.182** | 0.096 | 0.196 | 0.166 | 0.271 | 0.102 | 0.201 | 0.117 | 0.226 | 0.115 | 0.219 | 0.141 | 0.243 | 0.141 | 0.238 | 0.224 | 0.281 | 0.248 | 0.353 |
| | 48 | **0.127** | **0.212** | 0.149 | 0.244 | 0.331 | 0.391 | 0.158 | 0.248 | 0.169 | 0.268 | 0.186 | 0.235 | 0.211 | 0.300 | 0.198 | 0.283 | 0.321 | 0.354 | 0.440 | 0.470 |
| | 96 | **0.198** | **0.250** | 0.253 | 0.309 | 0.653 | 0.551 | 0.366 | 0.377 | 0.233 | 0.306 | 0.221 | 0.267 | 0.364 | 0.387 | 0.320 | 0.351 | 0.408 | 0.417 | 0.674 | 0.565 |
| | Avg | **0.121** | **0.202** | 0.142 | 0.229 | 0.312 | 0.354 | 0.175 | 0.250 | 0.150 | 0.246 | 0.150 | 0.226 | 0.205 | 0.285 | 0.193 | 0.271 | 0.280 | 0.321 | 0.379 | 0.416 |
| Avg of all | | **0.242** | **0.277** | 0.248 | 0.287 | 0.328 | 0.350 | 0.256 | 0.294 | 0.261 | 0.301 | 0.261 | 0.300 | 0.296 | 0.326 | 0.293 | 0.320 | 0.303 | 0.334 | 0.377 | 0.395 |

Table 14: Compared PCMLP with time series foundation models.

| Dataset | PCMLP (Ours) | | Timer-XL (2025a) | | Timer (2024b) | |
|---|---|---|---|---|---|---|
| | MSE | MAE | MSE | MAE | MSE | MAE |
| ECL | **0.132** | **0.223** | 0.138 | 0.233 | 0.159 | 0.244 |
| ETTh1 | **0.370** | **0.391** | 0.381 | 0.399 | 0.386 | 0.401 |
| Traffic | 0.396 | **0.244** | **0.387** | 0.260 | 0.413 | 0.265 |
| Weather | **0.151** | **0.191** | 0.165 | 0.209 | 0.176 | 0.215 |
| Solar-Energy | **0.179** | **0.225** | 0.200 | 0.229 | 0.204 | 0.234 |
| Average | **0.246** | **0.255** | 0.254 | 0.266 | 0.268 | 0.272 |

Table 15: Full ablation results with prediction lengths $H \in \{12, 24, 48, 96\}$ for PEMS and $H \in \{96, 192, 336, 720\}$ for others. The lookback length $L = 96$ for all experiments. Results are averaged over three runs with fixed seeds $\{0, 1, 2\}$. The best averaged results are in **red**.

| Setup | | ① Group | | ② Uniform | | ③ Linear | | ④ Add | | ⑤ w/o VAT | | ⑥ w/o TE | | ⑦ w/o Norm | | ⑧ Var-wise | |
|---|---|---|---|---|---|---|---|---|---|---|---|---|---|---|---|---|---|
| Metric | | MSE | MAE | MSE | MAE | MSE | MAE | MSE | MAE | MSE | MAE | MSE | MAE | MSE | MAE | MSE | MAE |
| ETTh1 | 96 | 0.370 | 0.391 | 0.375 | 0.395 | 0.371 | 0.393 | 0.685 | 0.542 | 0.377 | 0.395 | 0.377 | 0.395 | 0.383 | 0.405 | 0.381 | 0.396 |
| | 192 | 0.431 | 0.424 | 0.433 | 0.425 | 0.434 | 0.427 | 0.705 | 0.565 | 0.437 | 0.425 | 0.437 | 0.425 | 0.456 | 0.452 | 0.443 | 0.426 |
| | 336 | 0.482 | 0.446 | 0.482 | 0.446 | 0.480 | 0.445 | 0.714 | 0.580 | 0.480 | 0.446 | 0.481 | 0.446 | 0.502 | 0.469 | 0.485 | 0.448 |
| | 720 | 0.503 | 0.474 | 0.551 | 0.498 | 0.492 | 0.469 | 0.763 | 0.627 | 0.523 | 0.486 | 0.493 | 0.469 | 0.577 | 0.530 | 0.519 | 0.482 |
| | Avg | 0.447 | 0.434 | 0.460 | 0.441 | **0.444** | **0.433** | 0.717 | 0.579 | 0.454 | 0.438 | 0.447 | 0.434 | 0.480 | 0.464 | 0.457 | 0.438 |
| ETTh2 | 96 | 0.293 | 0.341 | 0.300 | 0.346 | 0.296 | 0.343 | 0.357 | 0.391 | 0.296 | 0.343 | 0.294 | 0.342 | 0.352 | 0.399 | 0.296 | 0.343 |
| | 192 | 0.368 | 0.391 | 0.370 | 0.391 | 0.378 | 0.394 | 0.424 | 0.429 | 0.371 | 0.391 | 0.373 | 0.391 | 0.432 | 0.444 | 0.371 | 0.392 |
| | 336 | 0.414 | 0.428 | 0.427 | 0.432 | 0.431 | 0.433 | 0.460 | 0.458 | 0.423 | 0.429 | 0.422 | 0.429 | 0.556 | 0.523 | 0.419 | 0.428 |
| | 720 | 0.434 | 0.446 | 0.437 | 0.447 | 0.438 | 0.448 | 0.461 | 0.466 | 0.435 | 0.447 | 0.430 | 0.444 | 0.933 | 0.699 | 0.433 | 0.447 |
| | Avg | **0.377** | **0.401** | 0.383 | 0.404 | 0.386 | 0.405 | 0.426 | 0.436 | 0.381 | 0.402 | 0.380 | 0.402 | 0.568 | 0.516 | 0.380 | 0.403 |
| ETTm1 | 96 | 0.305 | 0.344 | 0.308 | 0.347 | 0.306 | 0.344 | 0.679 | 0.532 | 0.313 | 0.352 | 0.321 | 0.355 | 0.310 | 0.358 | 0.325 | 0.358 |
| | 192 | 0.358 | 0.376 | 0.359 | 0.376 | 0.356 | 0.375 | 0.697 | 0.541 | 0.361 | 0.378 | 0.361 | 0.378 | 0.365 | 0.397 | 0.369 | 0.382 |
| | 336 | 0.386 | 0.395 | 0.387 | 0.397 | 0.385 | 0.395 | 0.706 | 0.549 | 0.391 | 0.399 | 0.393 | 0.400 | 0.396 | 0.422 | 0.402 | 0.404 |
| | 720 | 0.444 | 0.436 | 0.445 | 0.434 | 0.445 | 0.437 | 0.724 | 0.563 | 0.458 | 0.440 | 0.455 | 0.438 | 0.471 | 0.470 | 0.465 | 0.443 |
| | Avg | **0.373** | **0.388** | 0.375 | **0.388** | **0.373** | **0.388** | 0.701 | 0.546 | 0.381 | 0.392 | 0.383 | 0.393 | 0.385 | 0.412 | 0.390 | 0.397 |
| ETTm2 | 96 | 0.166 | 0.247 | 0.169 | 0.252 | 0.166 | 0.247 | 0.216 | 0.301 | 0.174 | 0.256 | 0.176 | 0.257 | 0.204 | 0.299 | 0.179 | 0.260 |
| | 192 | 0.230 | 0.290 | 0.232 | 0.293 | 0.230 | 0.290 | 0.275 | 0.334 | 0.237 | 0.297 | 0.239 | 0.298 | 0.280 | 0.351 | 0.240 | 0.298 |
| | 336 | 0.287 | 0.328 | 0.291 | 0.331 | 0.287 | 0.328 | 0.334 | 0.367 | 0.294 | 0.334 | 0.294 | 0.334 | 0.414 | 0.429 | 0.297 | 0.336 |
| | 720 | 0.388 | 0.388 | 0.389 | 0.389 | 0.389 | 0.389 | 0.430 | 0.418 | 0.393 | 0.392 | 0.392 | 0.392 | 0.561 | 0.522 | 0.398 | 0.393 |
| | Avg | **0.268** | **0.313** | 0.270 | 0.316 | **0.268** | **0.313** | 0.314 | 0.355 | 0.275 | 0.320 | 0.276 | 0.320 | 0.365 | 0.400 | 0.279 | 0.322 |
| ECL | 96 | 0.132 | 0.223 | 0.133 | 0.224 | 0.132 | 0.223 | 0.201 | 0.302 | 0.145 | 0.233 | 0.147 | 0.235 | 0.130 | 0.224 | 0.151 | 0.238 |
| | 192 | 0.150 | 0.239 | 0.150 | 0.239 | 0.150 | 0.239 | 0.213 | 0.312 | 0.157 | 0.245 | 0.160 | 0.248 | 0.148 | 0.241 | 0.163 | 0.249 |
| | 336 | 0.167 | 0.257 | 0.166 | 0.257 | 0.167 | 0.257 | 0.226 | 0.323 | 0.173 | 0.262 | 0.177 | 0.265 | 0.169 | 0.261 | 0.179 | 0.266 |
| | 720 | 0.204 | 0.291 | 0.201 | 0.290 | 0.205 | 0.292 | 0.263 | 0.350 | 0.211 | 0.295 | 0.218 | 0.302 | 0.214 | 0.301 | 0.217 | 0.301 |
| | Avg | **0.163** | **0.252** | **0.163** | **0.252** | 0.164 | 0.253 | 0.226 | 0.322 | 0.172 | 0.258 | 0.175 | 0.262 | 0.165 | 0.257 | 0.178 | 0.264 |
| Traffic | 96 | 0.396 | 0.244 | 0.401 | 0.245 | 0.402 | 0.244 | 0.533 | 0.322 | 0.406 | 0.248 | 0.436 | 0.260 | 0.507 | 0.251 | 0.446 | 0.263 |
| | 192 | 0.411 | 0.253 | 0.418 | 0.254 | 0.415 | 0.253 | 0.547 | 0.328 | 0.424 | 0.256 | 0.444 | 0.266 | 0.527 | 0.261 | 0.465 | 0.277 |
| | 336 | 0.428 | 0.259 | 0.433 | 0.260 | 0.424 | 0.259 | 0.557 | 0.333 | 0.442 | 0.266 | 0.454 | 0.272 | 0.543 | 0.269 | 0.479 | 0.274 |
| | 720 | 0.466 | 0.279 | 0.466 | 0.280 | 0.463 | 0.279 | 0.585 | 0.346 | 0.477 | 0.281 | 0.491 | 0.293 | 0.583 | 0.296 | 0.509 | 0.294 |
| | Avg | **0.425** | **0.259** | 0.430 | 0.260 | 0.426 | **0.259** | 0.556 | 0.332 | 0.437 | 0.263 | 0.456 | 0.273 | 0.540 | 0.269 | 0.475 | 0.277 |
| Weather | 96 | 0.151 | 0.191 | 0.152 | 0.193 | 0.151 | 0.191 | 0.169 | 0.216 | 0.161 | 0.202 | 0.162 | 0.204 | 0.150 | 0.198 | 0.172 | 0.214 |
| | 192 | 0.200 | 0.238 | 0.202 | 0.240 | 0.200 | 0.238 | 0.218 | 0.257 | 0.209 | 0.247 | 0.209 | 0.247 | 0.198 | 0.250 | 0.217 | 0.253 |
| | 336 | 0.258 | 0.282 | 0.260 | 0.283 | 0.257 | 0.282 | 0.273 | 0.296 | 0.264 | 0.288 | 0.264 | 0.288 | 0.256 | 0.296 | 0.271 | 0.292 |
| | 720 | 0.342 | 0.339 | 0.341 | 0.339 | 0.341 | 0.338 | 0.352 | 0.346 | 0.345 | 0.340 | 0.345 | 0.341 | 0.337 | 0.352 | 0.348 | 0.343 |
| | Avg | 0.238 | 0.263 | 0.239 | 0.264 | 0.237 | **0.262** | 0.253 | 0.279 | 0.245 | 0.269 | 0.245 | 0.270 | **0.235** | 0.274 | 0.252 | 0.276 |
| PEMS03 | 12 | 0.062 | 0.162 | 0.063 | 0.163 | 0.061 | 0.161 | 0.119 | 0.230 | 0.064 | 0.165 | 0.067 | 0.169 | 0.062 | 0.162 | 0.073 | 0.176 |
| | 24 | 0.083 | 0.186 | 0.086 | 0.189 | 0.081 | 0.184 | 0.156 | 0.259 | 0.088 | 0.193 | 0.097 | 0.201 | 0.079 | 0.180 | 0.110 | 0.216 |
| | 48 | 0.126 | 0.224 | 0.135 | 0.232 | 0.125 | 0.223 | 0.214 | 0.302 | 0.144 | 0.240 | 0.164 | 0.259 | 0.109 | 0.205 | 0.189 | 0.283 |
| | 96 | 0.177 | 0.270 | 0.185 | 0.276 | 0.175 | 0.268 | 0.275 | 0.351 | 0.196 | 0.288 | 0.259 | 0.328 | 0.136 | 0.226 | 0.284 | 0.351 |
| | avg | 0.112 | 0.211 | 0.117 | 0.215 | 0.110 | 0.209 | 0.191 | 0.285 | 0.123 | 0.221 | 0.147 | 0.239 | **0.097** | **0.193** | 0.164 | 0.257 |
| PEMS04 | 12 | 0.068 | 0.167 | 0.072 | 0.172 | 0.068 | 0.165 | 0.101 | 0.213 | 0.075 | 0.177 | 0.077 | 0.179 | 0.070 | 0.169 | 0.085 | 0.191 |
| | 24 | 0.078 | 0.178 | 0.082 | 0.185 | 0.076 | 0.175 | 0.112 | 0.223 | 0.090 | 0.194 | 0.096 | 0.201 | 0.078 | 0.178 | 0.111 | 0.219 |
| | 48 | 0.091 | 0.192 | 0.097 | 0.201 | 0.088 | 0.188 | 0.124 | 0.236 | 0.106 | 0.212 | 0.128 | 0.231 | 0.091 | 0.192 | 0.152 | 0.257 |
| | 96 | 0.102 | 0.203 | 0.109 | 0.213 | 0.100 | 0.199 | 0.134 | 0.245 | 0.120 | 0.227 | 0.160 | 0.258 | 0.102 | 0.203 | 0.186 | 0.286 |
| | avg | 0.085 | 0.185 | 0.090 | 0.193 | **0.083** | **0.182** | 0.118 | 0.229 | 0.098 | 0.203 | 0.115 | 0.217 | 0.085 | 0.186 | 0.134 | 0.238 |
| PEMS07 | 12 | 0.052 | 0.142 | 0.054 | 0.146 | 0.051 | 0.141 | 0.093 | 0.199 | 0.058 | 0.153 | 0.061 | 0.156 | 0.052 | 0.142 | 0.067 | 0.165 |
| | 24 | 0.062 | 0.154 | 0.066 | 0.161 | 0.061 | 0.152 | 0.101 | 0.205 | 0.075 | 0.175 | 0.084 | 0.182 | 0.062 | 0.154 | 0.096 | 0.197 |
| | 48 | 0.077 | 0.169 | 0.081 | 0.177 | 0.074 | 0.165 | 0.112 | 0.214 | 0.101 | 0.203 | 0.126 | 0.219 | 0.077 | 0.169 | 0.148 | 0.242 |
| | 96 | 0.097 | 0.184 | 0.098 | 0.191 | 0.093 | 0.178 | 0.135 | 0.229 | 0.128 | 0.229 | 0.172 | 0.253 | 0.097 | 0.184 | 0.204 | 0.283 |
| | avg | 0.072 | 0.162 | 0.075 | 0.169 | **0.070** | **0.159** | 0.110 | 0.212 | 0.091 | 0.190 | 0.111 | 0.203 | 0.072 | 0.162 | 0.129 | 0.222 |
| PEMS08 | 12 | 0.069 | 0.162 | 0.070 | 0.164 | 0.068 | 0.162 | 0.118 | 0.222 | 0.074 | 0.168 | 0.081 | 0.178 | 0.158 | 0.208 | 0.087 | 0.183 |
| | 24 | 0.089 | 0.182 | 0.091 | 0.185 | 0.088 | 0.180 | 0.149 | 0.246 | 0.103 | 0.197 | 0.119 | 0.214 | 0.151 | 0.180 | 0.131 | 0.224 |
| | 48 | 0.127 | 0.212 | 0.132 | 0.217 | 0.125 | 0.211 | 0.208 | 0.284 | 0.163 | 0.247 | 0.203 | 0.275 | 0.170 | 0.197 | 0.225 | 0.289 |
| | 96 | 0.198 | 0.250 | 0.201 | 0.251 | 0.201 | 0.250 | 0.316 | 0.339 | 0.265 | 0.311 | 0.342 | 0.347 | 0.190 | 0.210 | 0.355 | 0.360 |
| | avg | **0.121** | 0.202 | 0.124 | 0.204 | **0.121** | **0.201** | 0.198 | 0.273 | 0.151 | 0.231 | 0.186 | 0.253 | 0.167 | 0.199 | 0.200 | 0.264 |

performance of *PatchCat vs. the variable-wise tokenizer*. Our method ⑨ outperforms the variable-wise tokenizer ⑧ in all cases, with a mean MSE reduction of 21.9%.

## E  MORE RESULTS ABOUT VARYING INPUT LENGTH

We report results under varying input lengths. For a fair comparison, we use both MLP and Transformer backbones for variable-wise tokenizer (Var-wise) and PatchCat. As shown in Table 16, these experimental results confirm that PatchCat maintains its performance advantages across a wide range of context lengths.

Table 16: Comparing PatchCat with the variable-wise tokenizer (Var-wise) under varying input lengths $L$. PatchCat achieves better results across all lengths, including best length.

| Backbone | Setting | ECL | | | | | PEMS08 | | | | |
|---|---|---|---|---|---|---|---|---|---|---|---|
| | | $L$=96 | $L$=192 | $L$=384 | $L$=768 | Best | $L$=96 | $L$=192 | $L$=384 | $L$=768 | Best |
| MLP | Var-wise | 0.152 | 0.132 | 0.129 | 0.130 | 0.129 | 0.086 | 0.095 | 0.074 | 0.086 | 0.074 |
| | PatchCat | 0.132 | 0.127 | 0.127 | 0.128 | 0.127 | 0.069 | 0.069 | 0.074 | 0.076 | 0.069 |
| | Improve | 12.93% | 3.71% | 2.09% | 1.55% | 2.09% | 19.82% | 27.51% | -0.07% | 11.07% | 7.25% |
| Transformer | Var-wise | 0.149 | 0.134 | 0.133 | 0.135 | 0.133 | 0.085 | 0.086 | 0.080 | 0.086 | 0.080 |
| | PatchCat | 0.134 | 0.129 | 0.130 | 0.132 | 0.129 | 0.072 | 0.070 | 0.068 | 0.077 | 0.068 |
| | Improve | 9.98% | 3.66% | 2.61% | 1.95% | 2.80% | 15.53% | 18.66% | 15.23% | 10.03% | 15.23% |

## F  VISUALIZATION

We selected samples that met the criteria for *temporal mutations* from the PEMS08 dataset. As can be clearly seen from the Figure 9, PatchCat can accurately detect abrupt changes in the immediate future, thus correctly predicting future trends in the time series.

## G  THE USE OF LARGE LANGUAGE MODELS (LLMS)

This work does not involve LLMs as any important, original, or non-standard components. This work just uses LLMs to edit and improve the quality of existing text.

Figure 9: Visualization of temporal mutation samples.

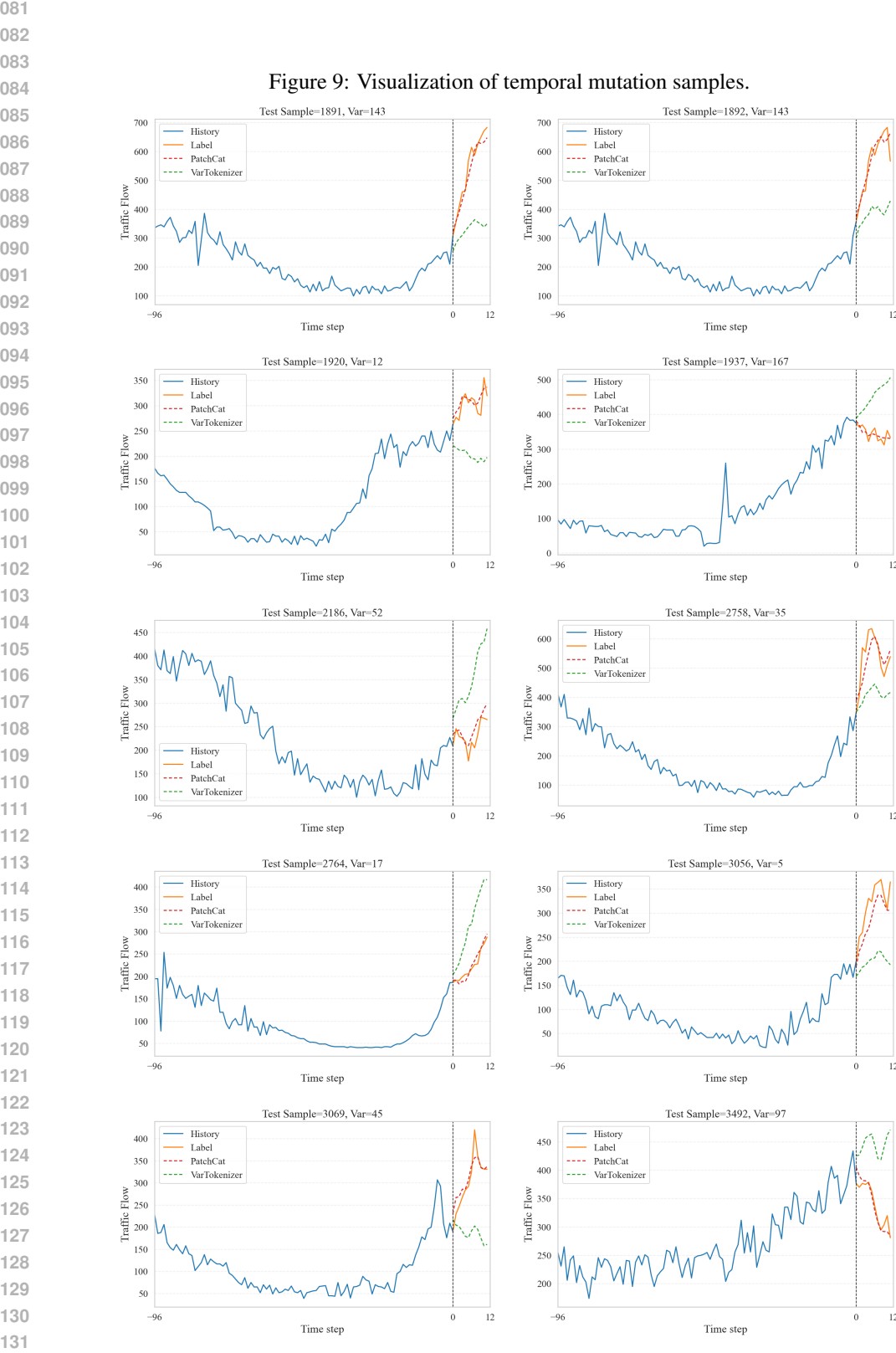