# OpenReview forum: "PatchCat: Rethinking Temporal Tokenization in Time Series Forecasting"
_ICLR.cc/2026/Conference — ICLR 2026 Conference Withdrawn Submission_

### Official Review · Reviewer_aGXL · 2025-11-01

**Soundness:** 2
**Presentation:** 2
**Contribution:** 2
**Rating:** 4
**Confidence:** 5

**Summary:**

The paper introduces PatchCat, a simple tokenization scheme that compresses each variable’s patch embeddings into a single token to balance accuracy and efficiency in time series forecasting. Using a lightweight MLP backbone (PCMLP), the method achieves strong results across multiple datasets with solid ablation and integration studies.

**Strengths:**

* The paper tackles a meaningful trade-off in time series forecasting, namely the accuracy of patch-wise tokenizers versus the efficiency of variable-wise ones. The motivation to balance these two aspects is sound.
* The proposed patch-then-concat strategy is easy to understand and implement. Compressing each variable’s patch information into a single token offers a clean bridge between existing tokenization paradigms.
* The experiments on multiple real-world datasets show that the proposed method achieves strong or even state-of-the-art results

**Weaknesses:**

* The name “PatchCat” and the method’s focus are very close to TimeCAT (ICLR 2025 submission), which also studies efficient, patch-based modeling for time series. The absence of any citation or comparison to this concurrent work makes the literature review feel incomplete.
* The paper claims that the “patch-then-concat” design preserves sequential information, but by collapsing all patch embeddings into a single token per variable, temporal relationships appear lost before reaching the backbone. In the PCMLP setting, an MLP cannot explicitly model temporal dependencies, making the approach resemble variable-wise tokenization rather than a genuine temporal one.
* The “linearly increasing dimension” rule is based on a simple “recent matters more” idea but lacks theoretical or empirical depth.

**Questions:**

* Please clarify the connection to TimeCAT and similar patch-based models. Given the clear overlap in motivation, a direct comparison or citation seems necessary.
* If all patches are compressed into a single token, how does PCMLP actually exploit sequential information? Can you provide evidence that temporal cues are retained and used effectively?
* Could you provide a stronger rationale or experimental support for the linear dimension allocation strategy? Why not use a nonlinear or learnable approach?

---

> ### Author Response · Authors · 2025-11-26
>
> ## Q1: Connection with TimeCAT
>
> TimeCAT is an excellent contribution to time‑series forecasting, and we have added proper citation in the revised manuscript. However, PatchCat and TimeCAT are conceptually different in several key aspects:
> - **Different meanings of “CAT”:**
> In TimeCAT, “CAT” stands for _Context‑Aware Transformer_, whereas in PatchCat it refers to _Concatenation_. The overlap in naming is purely coincidental.
> - **Different tokenization strategies:**
> TimeCAT converts the input series into a **sequence of tokens** using a patch‑based tokenizer with positional embeddings (Line 160 in the TimeCAT paper).
> In contrast, PCMLP uses PatchCat to **concatenate multiple patch embeddings into a single token** that represents the entire time series, leading to a fundamentally different modeling paradigm.
> - **Different purposes of concatenation:**
> TimeCAT’s Intra‑Group Operation (IGO) concatenates group tokens and value tokens to reduce computational cost, and the resulting representation is later split back into multiple tokens. PatchCat concatenates patches to preserve temporal order within one token. Moreover, PatchCat incorporates a linear dimension‑allocation strategy to better handle temporal mutation (new result in Table 6 and Figure 9 of the revised manuscript), a challenge not addressed in TimeCAT.
>
> ## Q2: Retaining and using sequential information
>
> PatchCat does **not** compress multiple patches into a single embedding. Instead, each patch is projected into a **small‑dimensional embedding** (d = 16 or higher), and these embeddings are then **concatenated in temporal order** to form a single token (typically d ≈ 528). This design explicitly preserves temporal order information inside the token.
>
> To address the reviewer’s concern, we add experiments analyzing temporal‑order sensitivity. When we reverse the temporal order of the input during testing, PatchCat shows a substantially larger performance degradation than variable‑wise tokenizers. This indicates that PatchCat indeed relies on the correct internal temporal ordering encoded within the concatenated token.
>
> | Setting      | ECL   |         |         | PEMS08 |         |         |
> | ------------ | ----- | ------- | ------- | ------ | ------- | ------- |
> |              | Ori   | Half-EX | Shuffle | Ori    | Half-EX | Shuffle |
> | PatchCat     | 0.132 | 0.256   | 1.003   | 0.069  | 1.495   | 1.610   |
> | VarTokenizer | 0.152 | 0.204   | 0.950   | 0.086  | 0.256   | 0.960   |
>
> ## Q3: Dimension allocation strategy
>
> **Comparison Experiment of Different Strategies**
> We evaluated three different dimension‑allocation strategies (group, uniform, linear) in Table 3 of the original manuscript.
>
> **The role of dimension allocation strategy**
> In the revised manuscript, We further discussed the essential role of dimension allocation strategies. We define _temporal mutation_ as a local trend reversal within the latest K steps. PatchCat responds more effectively to such mutations by allocating more embedding dimensions to recent patches. **New results (or see Table 6 in the revised manuscript)** show that PatchCat achieves roughly a 15% MSE reduction compared to variable‑wise tokenizers on samples exhibiting temporal mutations. **More interesting visualizations can be seen in Appendix F.**
>
> | Setting      | ECL    |        |        | PEMS08 |        |        |
> | ------------ | ------ | ------ | ------ | ------ | ------ | ------ |
> |              | K=3    | K=6    | K=12   | K=3    | K=6    | K=12   |
> | VarTokenizer | 0.153  | 0.154  | 0.151  | 0.089  | 0.087  | 0.086  |
> | PatchCat     | 0.133  | 0.134  | 0.131  | 0.074  | 0.072  | 0.071  |
> | Improve      | 13.36% | 13.24% | 13.12% | 17.42% | 17.23% | 16.90% |

---

### Official Review · Reviewer_QKem · 2025-11-02

**Soundness:** 2
**Presentation:** 3
**Contribution:** 2
**Rating:** 4
**Confidence:** 4

**Summary:**

This paper introduces PatchCat, which divides input time series into consecutive segments and concatenates their embeddings in chronological order. This approach preserves local semantics and sequential information while compressing univariate series into a single token, achieving efficiency similar to variable-wise processing. Besides, PatchCat employs a strategy of linearly increasing dimension allocation and applies variable-specific affine transformations to enhance its representational power. Experiments show that replacing the tokenizers in existing methods with PatchCat improves prediction performance.

**Strengths:**

1. This work tackles key technical issues in current temporal tokenizers by first categorizing them into three types: point-wise, patch-wise, and variable-wise. It then introduces PatchCat, a new approach that uses a "patch-then-concat" strategy to address the common trade-off between accuracy and efficiency.
2. The proposed design choices are closely aligned with the nature of time series data. Concretely, allocating dimensions in a linearly increasing manner is shown to be more effective than uniform allocation, supporting the idea that "recent events matter more." Besides, variable-wise affine transformation improves the model's ability to distinguish between different variables during tokenization.
3. Extensive experiments show the effectiveness of the approach. PCMLP achieves better performance than 8 state-of-the-art models across 11 real-world datasets. Ablation studies confirm the contribution of each component.

**Weaknesses:**

1. Core ideas of PatchCat and PCMLP  are incremental improvements, lacking original tokenization paradigms or architectural designs. .
2. This paper also suffers from weak motivation and insufficient visualization. No intuitive visualizations to highlight existing tokenizers’ limitations, and theoretical basis for key designs (e.g., linear dimension allocation) is insufficient.

**Questions:**

1. What are the innovations of the PCMLP structure compared with other methods? It seems to be just a stack of relatively simple modules.
2. In the results of Table 3, after integrating PatchCat, no significant performance improvement is achieved, which fails to accurately reflect the superiority of the proposed method.
3. The idea of combining the advantages of two types of temporal tokenizers is somewhat straightforward and common. Is it possible to emphasize the innovation?

---

> ### Author Response · Authors · 2025-11-26
>
> ## Motivation & Innovation & Visualization
>
> Although PatchCat and PCMLP adopt simple structures, their simplicity is precisely what enables the **simultaneous gains in both speed and accuracy** demonstrated in Figure 1. Importantly, the improvements are not accidental—we identify two concrete mechanisms behind PatchCat’s effectiveness in the revised manuscript:
> - **Preserve order information**: By concatenating multiple patch embeddings, PatchCat preserves temporal order information within one token. **New results (or see Table 5 in the revised manuscript)** show that reversing inputs leads to a significant performance drop, indirectly proving the above point.
>
> | Setting      | ECL   |         |         | PEMS08 |         |         |
> | ------------ | ----- | ------- | ------- | ------ | ------- | ------- |
> |              | Ori   | Half-EX | Shuffle | Ori    | Half-EX | Shuffle |
> | PatchCat     | 0.132 | 0.256   | 1.003   | 0.069  | 1.495   | 1.610   |
> | VarTokenizer | 0.152 | 0.204   | 0.950   | 0.086  | 0.256   | 0.960   |
>
> - **Handle temporal mutation**: We define _temporal mutation_ as a local trend reversal within the latest K steps. PatchCat responds more effectively to such mutations by allocating more embedding dimensions to recent patches. **New results (or see Table 6 in the revised manuscript)** show that PatchCat achieves roughly a 15% MSE reduction compared to variable‑wise tokenizers on samples exhibiting temporal mutations. **More interesting visualizations can be seen in Appendix F.**
>
> | Setting      | ECL    |        |        | PEMS08 |        |        |
> | ------------ | ------ | ------ | ------ | ------ | ------ | ------ |
> |              | K=3    | K=6    | K=12   | K=3    | K=6    | K=12   |
> | VarTokenizer | 0.153  | 0.154  | 0.151  | 0.089  | 0.087  | 0.086  |
> | PatchCat     | 0.133  | 0.134  | 0.131  | 0.074  | 0.072  | 0.071  |
> | Improve      | 13.36% | 13.24% | 13.12% | 17.42% | 17.23% | 16.90% |
>
> Regarding **PCMLP**, although its components are simple, its design intentionally leverages PatchCat’s strong single‑token embedding. By eliminating the need for complex temporal modeling layers, PCMLP achieves competitive accuracy with substantially lower computational cost.
>
> In summary, the revised manuscript not only validates PatchCat’s effectiveness but also identifies and analyzes its two key innovations—temporal‑information preservation and temporal‑mutation handling. These insights highlight PatchCat as a simple, reasonable, and efficient tokenizer design.
>
> ## No significant performance
>
> We acknowledge that the earlier presentation in the original manuscript, Table 3, may not have clearly conveyed the effectiveness of PatchCat, which may have led to misunderstanding. We provide a table reporting the percentage improvements. Across 24 cases (3 datasets × 4 baselines × 2 metrics), PatchCat achieves an average performance gain of **5.45%**, indicating that the improvements are consistent and non‑trivial.
>
> | Baseline | iTransformer |       | MMK   |       | DLinear |       | PatchTST |        |
> | -------- | ------------ | ----- | ----- | ----- | ------- | ----- | -------- | ------ |
> | Metric   | MSE          | MAE   | MSE   | MAE   | MSE     | MAE   | MSE      | MAE    |
> | ETTh1    | 2.65%        | 2.47% | 0.94% | 0.98% | 2.83%   | 3.79% | 5.74%    | 4.00%  |
> | ECL      | 9.02%        | 5.78% | 7.36% | 3.61% | 3.62%   | 4.15% | 18.18%   | 10.30% |
> | Traffic  | 2.10%        | 7.75% | 5.58% | 8.68% | 1.33%   | 4.21% | 7.62%    | 8.24%  |
> | Avg      | 4.59%        | 5.34% | 4.63% | 4.42% | 2.59%   | 4.05% | 10.51%   | 7.51%  |

---

### Official Review · Reviewer_7e9X · 2025-11-08

**Soundness:** 2
**Presentation:** 2
**Contribution:** 1
**Rating:** 2
**Confidence:** 4

**Summary:**

The paper proposes PatchCat, a method for tokenizing time series for forecasting models. Time series are split into patches and embedded with linearly increasing embedding dimension (recent patches have larger embedding size). The embeddings are then concatenated to construct a single token. Authors also proposed a complete model to validate this approach, which uses an MLP to map the token embedding into forecasts. Experiments have been conducted on 11 datasets showing improved results with the proposed approach.

**Strengths:**

- Authors explore a simple idea which works well, at least in the context of the results reported in this paper. Note: the empirical analysis has limitations, please see weaknesses for details.

**Weaknesses:**

- Although the paper investigates an interesting subproblem and demonstrates improved results, the main idea explored in the paper is not substantial enough for a main conference paper. As such, this work is better suited for a more focused venue (e.g., a time series workshop).
- The literature review only focuses on the relatively narrow long-term forecasting-style literature and makes no mention of time series foundation models, where the notion of tokenization has been explored from multiple angles and likely holds more significance than task-specific models. Relevant works (non-exhaustive list):
    - Ansari, Abdul Fatir, Stella, Lorenzo, et al. "Chronos: Learning the language of time series." arXiv preprint arXiv:2403.07815 (2024).
    - Woo, Gerald, et al. "Unified training of universal time series forecasting transformers." (2024): 53140.
    - Masserano, Luca, et al. "Enhancing foundation models for time series forecasting via Wavelet-based tokenization." arXiv preprint arXiv:2412.05244 (2024).
    - Talukder, Sabera, Yisong Yue, and Georgia Gkioxari. "Totem: Tokenized time series embeddings for general time series analysis." arXiv preprint arXiv:2402.16412 (2024).
    - Götz, Leon, et al. "Byte Pair Encoding for Efficient Time Series Forecasting." arXiv preprint arXiv:2505.14411 (2025).
    - Wang, Yihang, et al. "LightGTS: A Lightweight General Time Series Forecasting Model." arXiv preprint arXiv:2506.06005 (2025).
- The experiments mainly focus on scenarios with a fixed context length of 96. However, this is hardly the case in practice where one encounters varying series lengths. Moreover, for high frequency data (< 5min granularity), one would need context lengths of several thousand timesteps to correctly model seasonalities (e.g., weekly). In this case, the "token embedding" would grow dramatically which raises concerns about the generalizability of the method. I also have reservations considering this method as a type of tokenization for time series. Tokenization by its name implies something dynamic (a sequence), whereas this method just concatenates the embeddings into a single "token".
- The type of long-term forecasting benchmark considered in this work has been often criticized for its limitations. Please refer to the talk (and paper) from C. Bergmeir [1, 2] where he discusses the limitation of this benchmark and current evaluation practices. A recent position paper [3] also conducted a comprehensive evaluation of models on this benchmark showing that there's no obvious winner. Authors should consider using better benchmarks to demonstrate the effectiveness of their method. See, for example,
    - Chronos Benchmark II: This benchmark includes 27 datasets (42, if you include Benchmark I) providing a comprehensive coverage over domains, frequencies and other properties [4].
    - GIFT-Eval: This benchmark includes 90+ tasks across multiple datasets and domains. Please refer to https://github.com/SalesforceAIResearch/gift-eval.
    - The Monash Benchmark: https://forecastingdata.org/
    - or, the new released comprehensive fev-bench [5].

[1] https://neurips.cc/virtual/2024/workshop/84712#collapse108471
[2] Hewamalage, Hansika, Klaus Ackermann, and Christoph Bergmeir. "Forecast evaluation for data scientists: common pitfalls and best practices." Data Mining and Knowledge Discovery 37.2 (2023): 788-832.
[3] Brigato, Lorenzo, et al. "Position: There are no Champions in Long-Term Time Series Forecasting." arXiv preprint arXiv:2502.14045 (2025).
[4] Ansari, Abdul Fatir, et al. "Chronos: Learning the language of time series." arXiv preprint arXiv:2403.07815 (2024).
[5] Shchur, Oleksandr, et al. "fev-bench: A Realistic Benchmark for Time Series Forecasting." arXiv preprint arXiv:2509.26468 (2025).

- Apart from the limitations of benchmark, there are also problems with the evaluation of baselines. Restricting context length to 96 does not ensure a "fair comparison". Some models work better with longer context lengths and, in practice, context length is not restricted and longer contexts are used whenever possible. A fair comparison would be to either provide longer contexts for model that work well with them or experiment with different context lengths and report the best results for each model.
- Authors integrate the proposed method into PatchTST but I don't understand what this means. PatchTST is a transformer-based sequence model. If the entire sequence is embedded into a single token, it does not remain a sequence model anymore.

Some discussion, which can be made more accurate.

> Previous patchwise tokenizers required overlap between neighbor patches to avoid semantic fragmentation caused

This is hardly the case. Most patch-based models (see the time series foundation model literature) use non-overlapping patches and it does not pose any problem.

> Notably, despite not using complex inter-variable relationship modeling modules like Attention or Transformer, PCMLP still excels in datasets with a large number of variables.

This just suggests that the benchmark datasets considered have limited multivariate structure to exploit, a phenomenon which has been discussed in the literature previously.

**Questions:**

See above.

---

> ### Author Response · Authors · 2025-11-26
>
> ## Q1: Contribution
>
> While the earlier manuscript may have had presentation limitations, we respectfully clarify that our contribution goes beyond an incremental improvement. Our proposed PatchCat explicitly preserves local semantic and global temporal order in a single token, simplifying the complexity of the downstream backbone.
>
> In the revised manuscript, we add new experiments and visualizations to demonstrate that PatchCat’s performance gains arise from:
> - **Preserve order information**: By concatenating multiple patch embeddings, PatchCat preserves temporal order information within one token. **New results (or see Table 5 in the revised manuscript)** show that reversing inputs leads to a significant performance drop, indirectly proving the above point.
>
> | Setting      | ECL   |         |         | PEMS08 |         |         |
> | ------------ | ----- | ------- | ------- | ------ | ------- | ------- | --- |
> |              | Ori   | Half-EX | Shuffle | Ori    | Half-EX | Shuffle |
> | PatchCat     | 0.132 | 0.256   | 1.003   | 0.069  | 1.495   | 1.610 |
> | VarTokenizer | 0.152 | 0.204   | 0.950   | 0.086  | 0.256   | 0.960 |
>
> - **Handle temporal mutation**: We define _temporal mutation_ as a local trend reversal within the latest K steps. PatchCat responds more effectively to such mutations by allocating more embedding dimensions to recent patches. **New results (or see Table 6 in the revised manuscript)** show that PatchCat achieves roughly a 15% MSE reduction compared to variable‑wise tokenizers on samples exhibiting temporal mutations. **More interesting visualizations can be seen in Appendix F.**
>
> | Setting      | ECL    |        |        | PEMS08 |        |        |
> | ------------ | ------ | ------ | ------ | ------ | ------ | ------ |
> |              | K=3    | K=6    | K=12   | K=3    | K=6    | K=12   |
> | VarTokenizer | 0.153  | 0.154  | 0.151  | 0.089  | 0.087  | 0.086  |
> | PatchCat     | 0.133  | 0.134  | 0.131  | 0.074  | 0.072  | 0.071  |
> | Improve      | 13.36% | 13.24% | 13.12% | 17.42% | 17.23% | 16.90% |
>
> In the revised manuscript, we not only validate the effectiveness of PatchCat and analyze the core reasons for its performance gains but also reveal that PatchCat addresses an important and under‑explored challenge (temporal mutation).
>
> ## Q2: More relevant works
>
> Time‑series foundation (TFM) models aim for zero‑ or few‑shot generalization across diverse domains, whereas long‑term forecasting models—including ours—focus on fully supervised learning using domain‑specific training data. We have updated the revised manuscript to more comprehensively cover the literature on time‑series foundation models and their tokenizers.
>
> Nevertheless, we **add comparisons with TFM** in Table 14 of the revised manuscript.  Following the results from TIMER‑XL [2], PCMLP achieves the best results in 9 out of 10 cases. Moreover, PCMLP is substantially more parameter‑efficient: on ETTh1, it uses only 2.8% of TIMER‑XL’s parameters (2.4M vs. 84M) while reducing MAE by 2.88%.
>
> | Dataset      | PCMLP MSE | PCMLP MAE | Timer-XL MSE | Timer-XL MAE | Timer MSE | Timer MAE |
> | ------------ | --------- | --------- | ------------ | ------------ | --------- | --------- |
> | ECL          | **0.132** | **0.223** | 0.138        | 0.233        | 0.159     | 0.244     |
> | ETTh1        | **0.370** | **0.391** | 0.381        | 0.399        | 0.386     | 0.401     |
> | Traffic      | 0.396     | **0.244** | **0.387**    | 0.260        | 0.413     | 0.265     |
> | Weather      | **0.151** | **0.191** | 0.165        | 0.209        | 0.176     | 0.215     |
> | Solar-Energy | **0.179** | **0.225** | 0.200        | 0.229        | 0.204     | 0.234     |
> | **Average**  | **0.246** | **0.255** | 0.254        | 0.266        | 0.268     | 0.272     |

---

> > ### Author Response · Authors · 2025-11-26
> >
> > ## Q3: Varying input length
> >
> > PatchCat handles different input lengths through zero‑padding or truncation, similar to tokenizers used in iTransformer, DLinear, and other single‑token embedding approaches. We acknowledge this limitation. However, converting a time series into a single token provides significant practical benefits: it greatly simplifies the downstream model architecture and yields high computational efficiency, which is one of the central motivations of this work.
> >
> > Despite the longer input horizon, PatchCat maintains stronger embedding capability than the variable-wise tokenizer. In the original manuscript (Figure 6), we analyzed the impact of input length on performance. In the revised manuscript, we further expanded experiments to four different input lengths in Table 16. PatchCat consistently outperforms the variable‑wise tokenizer, demonstrating that its effectiveness is not restricted to the fixed length.
> >
> > | Backbone    | Setting      | ECL    |       |       |       |       | PEMS08 |        |        |        |        |
> > | ----------- | ------------ | ------ | ----- | ----- | ----- | ----- | ------ | ------ | ------ | ------ | ------ |
> > |             |              | L=96   | L=192 | L=384 | L=768 | Best  | L=96   | L=192  | L=384  | L=768  | Best   |
> > | MLP         | VarTokenizer | 0.152  | 0.132 | 0.129 | 0.130 | 0.129 | 0.086  | 0.095  | 0.074  | 0.086  | 0.074  |
> > |             | PatchCat     | 0.132  | 0.127 | 0.127 | 0.128 | 0.127 | 0.069  | 0.069  | 0.074  | 0.076  | 0.069  |
> > |             | Improve      | 12.93% | 3.71% | 2.09% | 1.55% | 2.09% | 19.82% | 27.51% | -0.07% | 11.07% | 7.25%  |
> > | Transformer | VarTokenizer | 0.149  | 0.134 | 0.133 | 0.135 | 0.133 | 0.085  | 0.086  | 0.080  | 0.086  | 0.080  |
> > |             | PatchCat     | 0.134  | 0.130 | 0.130 | 0.132 | 0.130 | 0.072  | 0.070  | 0.068  | 0.077  | 0.068  |
> > |             | Improve      | 9.98%  | 3.53% | 2.61% | 1.95% | 2.67% | 15.53% | 18.66% | 15.23% | 10.03% | 15.23% |
> >
> > ## Q4: Limitations of the experiment
> >
> > ### The limitations of the long‑term forecasting benchmark
> > We appreciate the reviewers’ suggestion regarding additional benchmarks. However, these datasets are mainly designed for benchmarking zero-/few-shot capabilities of TFM, rather than for evaluating supervised forecasting models considered in our work. Most of the subsets contain very limited samples or only a single variable, making them incompatible with the input requirements (e.g., multivariate structure, horizon/length settings) of our baselines.
> >
> > Nonetheless, we **add two extra datasets** collected from real‑world scenarios to further evaluate generalization in the revised manuscript. In conclusion, we conduct experiments on a **total of 13 datasets**.
> >
> > | Mode   | PCMLP     |           | TQNet     |       | TimeXer |       | CycleNet |       | iTransformer |       | MSGNet |       | TimesNet |       | PatchTST |       | DLinear |       |
> > | -- | --- | -- | -- | -- | -- | -- | -- | -- | -- | -- | -- | -- | -- | -- | -- | -- | -- | -- |
> > | Metric | MSE  | MAE   | MSE   | MAE   | MSE   | MAE   | MSE    | MAE   | MSE  | MA  | MSE  | MAE   | MSE      | MAE   | MSE  | MAE   | MSE   | MAE   |
> > | Solar  | 0.201     | **0.238** | **0.198** | 0.256 | 0.237   | 0.302 | 0.210    | 0.261 | 0.233        | 0.262 | 0.263  | 0.292 | 0.301    | 0.319 | 0.270    | 0.307 | 0.330   | 0.401 |
> > | BJAQ   | **0.452** | **0.435** | 0.462     | 0.442 | 0.466   | 0.448 | 0.459    | 0.439 | 0.482        | 0.455 | 0.456  | 0.448 | 0.481    | 0.458 | 0.460    | 0.444 | 0.456   | 0.481 |
> >
> > ### The context length
> > While most prior work adopts a fixed input length of 96 as a standard task setting, we agree with the reviewer that real‑world applications benefit from varying context lengths. Therefore, we report the best results for four varying input lengths {96, 192, 384, 768}, comparing variable‑wise tokenizer and PatchCat using both MLP and Transformer backbones. These experiments confirm that PatchCat maintains its performance advantages with varying context lengths.
> >
> > | Backbone    | Setting      | ECL   | PEMS08 |
> > | ----------- | ------------ | ----- | ------ |
> > | MLP         | VarTokenizer | 0.129 | 0.074  |
> > |             | PatchCat     | 0.127 | 0.069  |
> > |             | Improve      | 2.09% | 7.25%  |
> > | Transformer | VarTokenizer | 0.133 | 0.080  |
> > |             | PatchCat     | 0.130 | 0.068  |
> > |             | Improve      | 2.67% | 15.23% |
> >
> > ### Integrating PatchCat into PatchTST
> > Integrating PatchCat into PatchTST represents an extreme case where the PatchTST's backbone processes only a single token per variable, limiting it to capturing intra‑token autocorrelation. This experiment verifies whether a single PatchCat token can preserve temporal information without requiring multiple sequential tokens. The results show that PatchCat’s token retains sufficient temporal-order information, enabling notable speedups while maintaining competitive performance.

---

### Official Review · Reviewer_Gcj4 · 2025-11-11

**Soundness:** 2
**Presentation:** 3
**Contribution:** 3
**Rating:** 4
**Confidence:** 5

**Summary:**

This paper proposes a new tokenization strategy for transformer-based time series forecasting models. There are 3 widely-used tokenization strategies: tokenizing every single timestep, tokenizing patches (initially used in PatchTST), and tokenizing the entire time series as a single patch (initially proposed in iTransformer), a.k.a the variable patching mechanism. This paper interpolates between the two extremes to propose PatchCat, where time series sub-sequences of equal length are mapped to hidden dimensions $d$, where $d$ increases as we move from the first to the last patch, linearly. All the embeddings are then concatenated. The authors demonstrate that their methods perform well on some widely used datasets.

**Strengths:**

I believe that the paper presents a intuitive and simple idea, which seems to work well on the compared datasets, against competitive baselines.

**Weaknesses:**

1. **Limited comparative scope.**
   The paper does not compare PatchCat against time series foundation models (e.g., Chronos-1 and 2, TimesFM, Tirex, MOMENT, etc.), which have explored a diversity of tokenization and patching strategies. Moreover, the authors state that:

> Direct comparison of the above three kinds of temporal tokenizers across existing works is confounded by the usage of heterogeneous backbones.

but do not provide any experiments to address this gap. Without such baselines and experiments, it is difficult to contextualize the contribution of PatchCat relative to the current state of the art. I would highly recommend that the authors (1) compare their methods to these state-of-the-art forecasting models, (2) compare a wider diversity of tokenization strategies (see [2] Table 9, for more details), and (3) compare their proposed patching strategy against a variety of backbones.

2. **Dataset and evaluation limitations.**
   The datasets used (e.g., ETT, Exchange, Weather) are known to have limited diversity and issues. A few new benchmarks such as `GIFT-Eval` and `fev-bench` were released to partially address some of these gaps. I would highly recommend that the authors supplement their results with findings from these benchmarks.

> Finally, we use PCMLP as a baseline to compare the performance of PatchCat vs. the variable-wise tokenizer.

Moreover, prior work such as [1] and Chronos-2 have also reported that multivariate modeling yields little or no benefit on these public datasets, which weakens the empirical claims around PatchCat’s improvements.

3. **Hyperparameter tuning transparency.**
   The paper lists search ranges for learning rate and hidden dimensions but does not specify the tuning procedure—e.g., whether tuning was performed on a held-out validation set or the test set. This raises concerns about potential overfitting or unfair comparisons.

4. **Interpretability of “semantic fragmentation.”**
   The paper states that previous patchwise tokenizers required overlapping patches “to avoid semantic fragmentation.” However, the notion of “semantic fragmentation” is not clearly defined or empirically demonstrated. It remains unclear how the proposed no-overlap design preserves or enhances temporal semantics.

5. **Marginal novelty of time encoding and dimension allocation strategies.**
   The introduced time encoding resembles mechanisms used in earlier architectures such as AutoFormer, and recent works have questioned the necessity of explicit time encoding. Similarly, the dimension allocation strategy is intriguing but not fully motivated or analyzed, particularly for periodic or long-range temporal dependencies. From Table 2, it is unclear if any of the differences are significant.

### References
1. Żukowska, Nina, et al. "Towards long-context time series foundation models." arXiv preprint arXiv:2409.13530 (2024).
2. Goswami, Mononito, et al. "Moment: A family of open time-series foundation models." arXiv preprint arXiv:2402.03885 (2024).

**Questions:**

1. How exactly is *semantic fragmentation* defined and measured in this work? Can the authors clarify what empirical evidence supports the claim that overlap-free patching mitigates or avoids it?

2. How were hyperparameters tuned? Was there a distinct validation set, or was tuning performed directly on the test data? Clarifying this is critical for assessing experimental rigor.

3. Was the primary goal to isolate the effect of patch tokenization within a fixed base model (e.g., comparing PatchCat vs. variable-wise tokenization under identical architectures)? If so, it would be helpful to state this explicitly and explain why foundation model baselines were excluded.

4. Could the authors elaborate on why the proposed time encoding is expected to help, given that some recent studies (e.g., AutoFormer, TimesFM) have found temporal embeddings to have limited impact?

5. Do the authors anticipate that PatchCat would generalize beyond the benchmark datasets—especially to large-scale, real-world, irregularly sampled time series, or other modeling architectures, especially those used for foundation models?

---

> ### Author Response · Authors · 2025-11-26
>
> ## Q1: Concept of semantic fragmentation
> We use _semantic fragmentation_ to describe the limitation of representing temporal semantics with fixed‑length patches. Our comment that patching with overlapping may better cope with such a phenomenon refers to the fact that compared with non‑overlapping patching, overlap patching generates more tokens for the same input time series, potentially providing richer context.
>
> We introduce this concept to illustrate that PatchCat with non-overlapping patching can reduce computational load without sacrificing performance. We believe this advantage stems from the concatenation of multiple patches, integrating semantics into a single token. We **add a row of new results in Table 2 of the revised manuscript** to demonstrate that overlapping does not outperform the non-overlapping mode.
>
> | Setting        | ETT(avg)  | ECL       | Traffic   | Weather   | PEMS03    | PEMS04    | PEMS07    | PEMS08    | Avg       |
> | -------------- | --------- | --------- | --------- | --------- | --------- | --------- | --------- | --------- | --------- |
> | Overlapping    | 0.371     | **0.164** | 0.427     | 0.238     | 0.113     | **0.082** | 0.073     | 0.122     | 0.199     |
> | No-overlapping | **0.368** | **0.164** | **0.426** | **0.237** | **0.110** | 0.083     | **0.070** | **0.121** | **0.197** |
>
> ## Q2: Commitment to rigorous hyperparameter search
> All hyperparameters were tuned solely on the training–validation splits of each dataset, using the validation set for model selection. The test set was never used during tuning and was only evaluated once after training. We have clarified this in the revised manuscript.
>
> ## Q3: Fair comparison
>
> ### Q3.1: Fair comparison using fixed-base model
> As the reviewer notes, our primary goal is indeed to isolate the effect of tokenization within a fixed model. In the original manuscript, we replaced the variable‑wise tokenizers in iTransformer, MMK, and DLinear with PatchCat (Table 3) and compared variable‑wise versus PatchCat in the ablation study (Table 2). Across both experiment groups, PatchCat consistently yields performance improvements.
>
> ### Q3.2: Comparison with time foundation models
> Time foundation models (TFM) were not included as baselines because they pursue a different objective—zero/few‑shot forecasting—while long‑term forecasting models, including ours, rely on supervised learning from domain‑specific data. Nevertheless, we **add comparisons with TFM** in Table 14 of the revised manuscript. Following the results from TIMER‑XL [1], PCMLP achieves the best results in 9 out of 10 cases. Moreover, PCMLP is substantially more parameter‑efficient: on ETTh1, it uses only 2.8% of TIMER‑XL’s parameters (2.4M vs. 84M) while reducing MAE by 2.88%.
>
> | Dataset      | PCMLP MSE | PCMLP MAE | Timer-XL MSE | Timer-XL MAE | Timer MSE | Timer MAE |
> | ------------ | --------- | --------- | ------------ | ------------ | --------- | --------- |
> | ECL          | **0.132** | **0.223** | 0.138        | 0.233        | 0.159     | 0.244     |
> | ETTh1        | **0.370** | **0.391** | 0.381        | 0.399        | 0.386     | 0.401     |
> | Traffic      | 0.396     | **0.244** | **0.387**    | 0.260        | 0.413     | 0.265     |
> | Weather      | **0.151** | **0.191** | 0.165        | 0.209        | 0.176     | 0.215     |
> | Solar-Energy | **0.179** | **0.225** | 0.200        | 0.229        | 0.204     | 0.234     |
> | **Average**  | **0.246** | **0.255** | 0.254        | 0.266        | 0.268     | 0.272     |
>
> ## Q4: The impact of temporal embedding
> Most existing studies adopt one of the following time‑encoding strategies: (1) one-hot encoding of time features (e.g., hour/minute/second), (2) RoPE‑style positional encodings, or (3) learnable temporal embeddings. Findings in STID [2] demonstrate that the learnable temporal embeddings can effectively capture periodic patterns.
>
> As stated in our original paper, we use the same time‑encoding strategy proposed in STID. We believe that this strategy enables the model to better utilize the periodicity in the data. While models such as AutoFormer and TimesFM are sufficiently complex to learn periodic patterns implicitly, our PCMLP architecture is intentionally simple, making the explicit periodic cues provided by learnable time encoding particularly beneficial.

---

> > ### Author Response · Authors · 2025-11-26
> >
> > ## Q5: Generalization
> >
> > We appreciate the reviewers’ suggestion regarding additional benchmarks. However, these datasets are mainly designed for benchmarking zero-/few-shot capabilities of TFM, rather than for evaluating supervised forecasting models considered in our work. Most of the subsets contain very limited samples or only a single variable, making them incompatible with the input requirements (e.g., multivariate structure, horizon/length settings) of our baselines.
> >
> > Nonetheless, we **add two extra datasets** collected from real‑world scenarios to further evaluate generalization in the revised manuscript. Extending PatchCat to irregularly sampled time series is an interesting direction that we plan to explore in future work.
> >
> > | Mode   | PCMLP     |           | TQNet     |       | TimeXer |       | CycleNet |       | iTransformer |       | MSGNet |       | TimesNet |       | PatchTST |       | DLinear |       |
> > | ------ | --------- | --------- | --------- | ----- | ------- | ----- | -------- | ----- | ------------ | ----- | ------ | ----- | -------- | ----- | -------- | ----- | ------- | ----- |
> > | Metric | MSE       | MAE       | MSE       | MAE   | MSE     | MAE   | MSE      | MAE   | MSE          | MAE   | MSE    | MAE   | MSE      | MAE   | MSE      | MAE   | MSE     | MAE   |
> > | Solar  | 0.201     | **0.238** | **0.198** | 0.256 | 0.237   | 0.302 | 0.210    | 0.261 | 0.233        | 0.262 | 0.263  | 0.292 | 0.301    | 0.319 | 0.270    | 0.307 | 0.330   | 0.401 |
> > | BJAQ   | **0.452** | **0.435** | 0.462     | 0.442 | 0.466   | 0.448 | 0.459    | 0.439 | 0.482        | 0.455 | 0.456  | 0.448 | 0.481    | 0.458 | 0.460    | 0.444 | 0.456   | 0.481 |
> >
> > ## Reference
> >
> > - [1] Liu Y, Qin G, Huang X, et al. Timer-xl: Long-context transformers for unified time series forecasting. The Thirteenth International Conference on Learning Representations.
> > - [2] Shao, Zezhi, et al. "Spatial-temporal identity: A simple yet effective baseline for multivariate time series forecasting." _Proceedings of the 31st ACM international conference on information & knowledge management_. 2022.

---

### Author Response · Authors · 2025-11-26

We sincerely thank all reviewers for their constructive and insightful feedback. We have substantially revised the manuscript with new analyses, experiments, and clearer positioning. Below we summarize the major updates.

**1. Clarifying the core contributions of PatchCat.**
We highlight two concrete and verifiable mechanisms underlying PatchCat:
(1) **Preserving temporal order.** By concatenating multiple patch embeddings, PatchCat preserves temporal order information within one token (**Table 5**).
(2) **Handling temporal mutation.** PatchCat responds more effectively to such mutations by allocating more embedding dimensions to recent patches (**Table 6, Fig. 9**).

**2. Broadening the evaluation of PCMLP.**
- We extend comparisons to Time Series Foundation Models. PCMLP outperforms TIMER‑XL in 9/10 cases while using only 2.8% of its parameters on ETTh1 (**Table 14**).
- To further assess generalization, we include two additional real‑world datasets, expanding evaluation to 13 datasets in total (**Table 1**).

**3. Additional experiments for rigor and fairness.**
- We add overlapping vs. non-overlapping patching experiments (**Table 2**) and observe no benefit from overlapping, supporting the efficiency of PatchCat’s non-overlapping design.
- To address concerns regarding fixed input lengths, we report results across four horizons {96, 192, 384, 768}, showing consistent improvements with both MLP and Transformer backbones (**Table 16**).

We have added 100+ new experimental results and visualizations in the rebuttal stage. We believe these extensive revisions significantly enhance the clarity, empirical support, and contribution of the work.

---

### Note · Authors · 2026-01-20

I have read and agree with the venue's withdrawal policy on behalf of myself and my co-authors.